# Quantification of T-cell dynamics during latent cytomegalovirus infection in humans

Sara P. H. van den Berg[1,2], Lyanne Y. Derksen[2], Julia Drylewicz[2], Nening M. Nanlohy[1], Lisa Beckers[1], Josien Lanfermeijer[1,2], Stephanie N. Gessel[1], Martijn Vos[1], Sigrid A. Otto[2], Rob J. de Boer[3], Kiki Tesselaar[2], José A. M. Borghans[2ᵒ], Debbie van Baarle[1,2ᵒ¤*]

1 Center for Infectious Disease Control, National Institute for Public Health and the Environment, Bilthoven, the Netherlands, 2 Center for Translational Immunology, University Medical Center Utrecht, Utrecht, the Netherlands, 3 Theoretical Biology, Utrecht University, Utrecht, the Netherlands

ᵒ These authors contributed equally to this work.
¤ Current address: Department of Medical Microbiology and Infection prevention, University Medical Center Groningen, Groningen, the Netherlands
* d.van.baarle@umcg.nl

**Data Availability Statement:** All relevant data are within the manuscript and its Supporting Information files.

## Abstract

Cytomegalovirus (CMV) infection has a major impact on the T-cell pool, which is thought to be associated with ageing of the immune system. The effect on the T-cell pool has been interpreted as an effect of CMV on non-CMV specific T-cells. However, it remains unclear whether the effect of CMV could simply be explained by the presence of large, immunodominant, CMV-specific memory CD8+ T-cell populations. These have been suggested to establish through gradual accumulation of long-lived cells. However, little is known about their maintenance. We investigated the effect of CMV infection on T-cell dynamics in healthy older adults, and aimed to unravel the mechanisms of maintenance of large numbers of CMV-specific CD8+ T-cells. We studied the expression of senescence, proliferation, and apoptosis markers and quantified the *in vivo* dynamics of CMV-specific and other memory T-cell populations using *in vivo* deuterium labelling. Increased expression of late-stage differentiation markers by CD8+ T-cells of CMV+ versus CMV- individuals was not solely explained by the presence of large, immunodominant CMV-specific CD8+ T-cell populations. The lifespans of circulating CMV-specific CD8+ T-cells did not differ significantly from those of bulk memory CD8+ T-cells, and the lifespans of bulk memory CD8+ T-cells did not differ significantly between CMV- and CMV+ individuals. Memory CD4+ T-cells of CMV+ individuals showed increased expression of late-stage differentiation markers and decreased Ki-67 expression. Overall, the expression of senescence markers on T-cell populations correlated positively with their expected *in vivo* lifespan. Together, this work suggests that i) large, immunodominant CMV-specific CD8+ T-cell populations do not explain the phenotypical differences between CMV+ and CMV- individuals, ii) CMV infection hardly affects the dynamics of the T-cell pool, and iii) large numbers of CMV-specific CD8+ T-cells are not due to longer lifespans of these cells.

**Funding:** This research was (partially) funded by the Netherlands Organisation for Scientific Research (NWO) Science domain NWO-ENW, project ALWOP.512 (https://www.nwo.nl/en/science-enw) (to JAMB), and the Strategic Program Research (SPR) of the National Institute of Public Health and the Environment (RIVM) (https://www.rivm.nl/en/about-rivm/mission-and-strategy/our-work/tasks/knowledge-development-and-research/strategic-research-0) (to DB). The funders had no role in study design, data collection and analysis, decision to publish, or preparation of the manuscript.

**Competing interests:** The authors have declared that no competing interests exist.

## Author summary

Most people are infected with cytomegalovirus (CMV). Once infected, CMV stays in our body for the rest of our life. It rarely causes severe disease, thanks to T-cells that keep the virus at bay. CMV induces exceptionally large T-cell numbers. While typically no more than 1% of our T-cells recognize a certain virus, for CMV this can be as much as 50%. CMV is therefore used in vaccines to induce high T-cell numbers to other viruses or bacteria. A possible downside of CMV is that it is thought to lead to faster ageing of the immune system. We have studied whether the exceptionally large number of CMV-specific T-cells is due to a lack of cell death of these cells, and whether CMV infection affects T-cells to other viruses. We found that the changes in the T-cell pool of CMV-infected individuals cannot be explained by the presence of large numbers of CMV-specific T-cells. This suggests that CMV infection may also affect T-cells specific for other viruses. We also found that CMV-specific T-cells live as long as T-cells specific for other viruses. These insights are important for our understanding of the effects of CMV infection and for the development of CMV-based vaccines.

## Introduction

In an ongoing fight against new and emerging viruses and bacteria, the adaptive immune system provides a unique line of defence for its host. Adaptive immunity is not only highly specific against pathogens, it also enables protection for many years. Immunological memory requires long-term maintenance of T-cell memory. After an infection is cleared, the pool of generated effector memory ($T_{EM}$) and central memory ($T_{CM}$) T-cells typically contracts, and a small population of antigen-specific T-cells is maintained over time. Under steady-state conditions, cell numbers are maintained by balancing the production of new cells, either from a source or from proliferation, with the loss of cells, either by death or differentiation. Although it is challenging to measure the lifespan of cell populations *in vivo* in humans, it is important to obtain such quantitative insights in order to understand how T-cell memory is maintained. Moreover, several features of the T-cell compartment can be severely impacted by chronic viral infections, specifically by human cytomegalovirus (CMV), which need to be taken into account.

Cytomegaloviruses are archaic double-stranded DNA viruses that have infected vertebrates for millennia [1]. CMV can cause pathology in severely immunocompromised states (e.g. after organ transplantation, or in advanced HIV infection), but in healthy individuals the primary infection typically remains asymptomatic. Nevertheless, CMV infection greatly influences the T-cell phenotype of healthy individuals—CD8$^+$ T-cells in particular. It leads to a marked increase in the absolute number of CD8$^+$ $T_{EM}$ [2] and effector memory re-expressing CD45RA ($T_{EMRA}$) T-cells [3,4]. These cells also seem functionally different, as they show decreased expression of the co-stimulatory molecules CD27 and CD28 [3,4], and upregulation of the senescence-associated markers CD57 and KLRG-1 [5]. Taken together, the effects of CMV on the T-cell pool resemble general age-associated changes [2]. This has prompted the hypothesis that CMV infection contributes to the age-related decay of immune function, including diminished responses to infectious diseases and vaccination [6,7]. An explanation for the CMV-induced changes to the T-cell pool is lacking. As much as 30%-90% of the total CD8$^+$ T-cell pool can be specific for CMV [8,9], with individual immunodominant CMV-specific CD8$^+$ T-cell responses frequently reaching 15%. Therefore, CMV-induced changes might–at least to

some extent–be explained by the presence of extremely large numbers of CMV-specific T-cells.

In mice, it has been shown that the number of CMV-specific T-cells, both $T_{CM}$ and $T_{EM/EMRA}$ cells, gradually increase over time, a process termed 'memory inflation' [10–15]. In humans, there is only some evidence for memory inflation [13,16–18], and *how* a handful of T-cell clones can be so numerous, or even increase over time, is not well understood. The underlying dynamics, i.e. the production and loss rates, of both CMV-specific T-cells and other memory T-cells in CMV-infected individuals, remain largely unknown. Based on *in vivo* deuterated glucose labelling of older adults (>65 years), it has previously been proposed that CMV-specific $CD8^+$ T-cells are relatively long-lived and thereby accumulate over time [19]. However, the only CMV-seropositive (CMV+) individual in a deuterated water labelling study [20] had higher $CD8^+$ bulk memory T-cell production rates than the four CMV- individuals, which could possibly be explained by chronic, age-related immune activation [21] as was suggested for mice [22]. As CMV has such a broad impact on the T-cell pool, it is important to understand the functional implications of and mechanisms underlying these CMV-induced changes.

In this study, we aimed to investigate the effect of human CMV infection on $CD4^+$ and $CD8^+$ T-cell dynamics in healthy older adults, as we expected that the effects of CMV would be the largest in older individuals, who have the highest chance of having been CMV-infected for a long time [2]. We aimed to unravel the mechanisms of maintenance of immunodominant CMV-specific $CD8^+$ T-cell responses. Hereto, we extensively characterized the lifespan-associated phenotype of different T-cell subsets with flow cytometry in a cohort of 32 CMV+ and 22 CMV- healthy older adults. In five CMV- and five CMV+ of these adults, we additionally performed a long-term deuterated water ($^2H_2O$) labelling study to assess the *in vivo* dynamics of cells. We show that the changes in the phenotype of the T-cell pool of CMV+ individuals are not solely explained by the presence of large, immunodominant CMV-specific $CD8^+$ T-cell populations. In terms of lifespan-associated phenotype, of all memory T-cell populations $CD4^+$ $T_{EM}$, $CD4^+$ $T_{EMRA}$, and $CD8^+$ $T_{EM}$ cells differ most between CMV+ and CMV- individuals, e.g. by increased expression of senescence markers and decreased Ki-67 expression in CMV+ individuals. Despite the observed differences in the expression of senescence markers and Ki-67, we found neither a significant difference in $CD8^+$ T-cell dynamics between CMV+ and CMV- individuals, nor between CMV-specific $CD8^+$ T-cells and bulk memory $CD8^+$ T-cells. Finally, the expression of senescence markers and Ki-67 by different T-cell populations correlated significantly with their *in vivo* production rates.

## Results

### Characterization of the study cohort

We included 54 healthy older adults of on average 68.8 years of age (SD 4.6 years, range 61.0 to 79.8 years) who were free of systemic diseases or immunocompromising conditions. Individuals with any (history of) immune-mediated diseases or any current medicine use (with the exception of occasional use of paracetamol or ibuprofen) were excluded. Using a multiplex immunoassay to test for CMV antibodies, we identified 32 of the 54 individuals (59%) as CMV+ and 22 as CMV- (S1 Table). No significant differences in sex or age were observed between CMV+ and CMV- individuals.

We investigated the influence of CMV-seropositivity on the T-cell pool by measuring absolute T-cell numbers, and their memory or senescence-associated phenotype in CMV- and CMV+ individuals. In line with previous reports [2,4,23,24], we found significantly increased cell numbers of $CD4^+$ and $CD8^+$ T-cells with a $T_{EM}$ or $T_{EMRA}$ phenotype in CMV+ compared to CMV- individuals (Figs 1A and S1, latter for gating strategy). CMV-seropositivity was

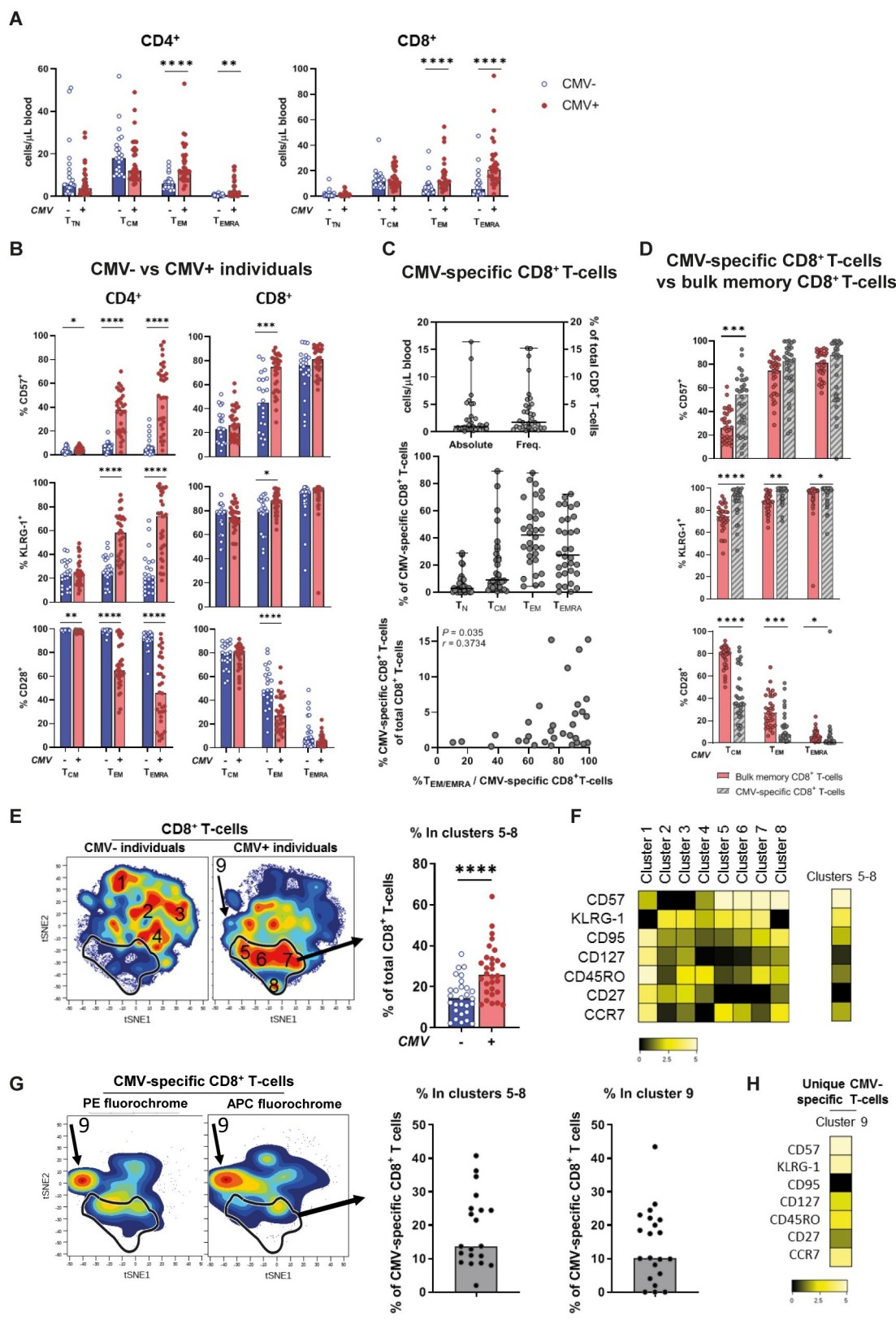

**Fig 1. Alterations in T-cell pool in CMV infection are not solely explained by the presence of CMV-specific CD8$^+$ T-cells. (A)** Absolute T-cell numbers in CMV- (in blue) and CMV+ (in red) individuals for CD4$^+$ (left panel) and CD8$^+$ (right panel) T$_{TN}$, T$_{CM}$, T$_{EM}$, and T$_{EMRA}$ cells. **(B)** Expression of CD57, KLRG-1, and CD28 by CD4$^+$ and CD8$^+$ T-cells, measured per memory subpopulation (T$_{CM}$, T$_{EM}$, and T$_{EMRA}$) and compared between CMV+ and CMV- individuals. **(C)** CMV-specific CD8$^+$ T-cells analyzed using 7 different HLA-class I dextramers for immunodominant CMV epitopes (yielding $N$ = 38 CMV-specific CD8$^+$ T-cell populations in $N$ = 32 CMV+ individuals–see S1 Table for epitopes used). Upper panel: CMV-specific CD8$^+$ T-cells in absolute numbers and frequency of total CD8$^+$ T-cells. Middle panel: phenotype of CMV-specific CD8$^+$ T-cells based on division in T$_N$ (naïve, CD27$^+$CD45RO$^-$), T$_{CM}$, T$_{EM}$, and T$_{EMRA}$ cells. Lower panel: association of percentage T$_{EM/EMRA}$ phenotype of CMV-specific CD8$^+$ T-cells and the frequency of CMV-specific cells within the CD8$^+$ T-cell pool. Error bars in upper and middle panel represent the range. **(D)** Comparison of expression of CD57, KLRG-1, and CD28 by CMV-specific CD8$^+$ T-cells and total memory CD8$^+$ T-cells of CMV+ individuals, analyzed per memory subpopulation (T$_{CM}$, T$_{EM}$, and T$_{EMRA}$). **(E)** Left panels: CD8$^+$ T-cells in CMV- and CMV+ individuals, clustered by t-SNE analysis based on the expression of CD57, KLRG-1, CD127, CD27, CD45RO, CCR7, and CD95. Numbers 1–9 represent identified clusters of CD8$^+$ T-cells based on cell density (with red representing high density of cells, and blue low density of cells). Cluster 5–8 (CD57$^{high}$, KLRG-1$^{high}$, CD27$^{low}$) are outlined in black in all t-SNE graphs. Right panels: Percentage of cluster 5–8 within the total CD8$^+$ T-cell pool of CMV- and CMV+ individuals. **(F)** Phenotype of cluster 5–8 is presented in a heatmap by the mean fluorescence intensity of CD57, KLRG-1, CD127, CD27, CD45RO, CCR7, and CD95 as used in the t-SNE analysis. **(G)** Left panel: CMV-specific CD8$^+$ T-cells clustered by the same t-SNE analysis as total CD8$^+$ T-cells. Middle panel: Percentage of clusters 5–8 among CMV-specific T-cells. Right panel: Percentage of cluster 9 among CMV-specific T-cells. **(H)** Phenotype of cluster 9 is presented in a heatmap by the mean fluorescence intensity of CD57, KLRG-1, CD127, CD27, CD45RO, CCR7, and CD95 as used in the t-SNE analysis. Expression levels are given in arbitrary units ranging from zero (black) to high (white). Bars and horizontal lines in all figures represent medians. Differences between CMV- and CMV+ individuals were assessed by Mann-Whitney $U$ test. Correlation was tested by Spearman's correlation. Stars indicate significant differences as follows: * $P$-value <0.05, ** $P$-value <0.01, *** $P$-value <0.001, *** $P$-value <0.0001.

associated with a significantly increased expression of the senescence markers CD57 and KLRG-1 and a loss of expression of the co-stimulatory receptor CD28 on CD4$^+$ T$_{EM}$, CD4$^+$ T$_{EMRA}$ and CD8$^+$ T$_{EM}$ cell populations, identifying a late-stage differentiation state (Fig 1B).

In addition, 38 CMV-specific CD8$^+$ T-cell populations were characterized in 32 CMV+ participants using HLA-class I dextramers for immunodominant CMV epitopes, based on the HLA-type of each participant (S1 Table). The frequencies of CMV-specific CD8$^+$ T-cells specific for one epitope reached on average 3.7% of total CD8$^+$ T-cells and could reach up to 15–20% for some epitopes (Fig 1C upper panel). The dominant phenotype of CMV-specific CD8$^+$ T-cells varied between individuals, although most had a T$_{EM}$/T$_{EMRA}$ phenotype (Fig 1C middle panel). We observed a significant positive correlation between the frequency of CMV-specific CD8$^+$ T-cells and the percentage of CMV-specific CD8$^+$ T-cells with a T$_{EM/EMRA}$ phenotype ($P$ = 0.035, $r$ = 0.37) (Fig 1C lower panel). Furthermore, even within the different memory subsets, CMV-specific CD8$^+$ T-cells showed a more pronounced late-stage differentiation state compared to bulk memory cells within each subset. This was most apparent in the T$_{CM}$ and T$_{EM}$ compartments, in which CMV-specific CD8$^+$ T-cells showed a higher expression of the markers CD57 and KLRG-1 and a lower expression of the receptor CD28 than bulk T$_{CM}$ and T$_{EM}$ cells (Fig 1D). Thus, CMV-specific CD8$^+$ T-cells were the most differentiated, followed by bulk memory CD8$^+$ T-cells in CMV+ individuals, and then bulk memory CD8$^+$ T-cells in CMV- individuals.

## Alterations in the CD8$^+$ T-cell pool in CMV infection are not solely explained by the presence of CMV-specific T-cells

Although the impact of a latent CMV infection on the T-cell pool is well known [2,3,5], the underlying mechanism for these changes remains unclear. Given that the altered phenotype of the CD8$^+$ T-cell pool of CMV+ individuals resembles the phenotype of immunodominant CMV-specific CD8$^+$ T-cells, we investigated whether this altered phenotype could be explained by the presence of large, immunodominant CMV-specific CD8$^+$ T-cell populations. To this end, we used a mean fluorescence intensity (MFI)-based t-Distributed Stochastic Neighbor Embedding (t-SNE) cluster analysis, including the markers CD57, KLRG-1, CD127, CD27, CD45RO, CCR7, and CD95. T-SNE analysis of CD8$^+$ T-cells from CMV- ($N$ = 22) and

CMV+ ($N$ = 32) individuals revealed large differences in T-cell phenotype (Fig 1E and 1F). CMV+ individuals had significantly more cells in clusters 5–8 ($P < 0.0001$) (Figs 1E and S2A). These clusters represent CD57$^{high}$KLRG-1$^{high}$CD27$^{low}$ T-cells (Fig 1F), whereas clusters 1–4 in CMV- individuals are characterized by either low KLRG-1 (cluster 1) or low CD57 (clusters 2–4) expression. Next, we plotted CMV-specific CD8$^+$ T-cells specific for different immunodominant CMV-epitopes on the same t-SNE cluster-axes (Fig 1G left panel). Only 13.7% (median, range 2.0–40.8%) of CMV-specific CD8$^+$ T-cells were part of clusters 5–8, the clusters that were increased in CMV+ individuals (Fig 1E middle panel). In addition, a proportion of CMV-specific CD8$^+$ T-cells contributed to cluster 9, a cluster that was hardly observed in CMV- individuals. CMV-specific CD8$^+$ T-cells in cluster 9 were observed in the majority of CMV+ individuals (S2B Fig) and consisted of 10.2% (median, range 0.0–43.5%) of CMV-specific CD8$^+$ T-cells per donor (Fig 1G right panel). The unique combination of T-cell markers in cluster 9 seems to be exclusive for CMV-specific CD8$^+$ T-cells, a phenotype that is not typically associated with CMV-specific CD8$^+$ T-cell responses [9]. This phenotype can be described as T$_{CM}$ or T$_{EM}$ like (CD27$^{medium}$CD45RO$^{high}$) with high expression of the senescence markers CD57 and KLRG-1, but low expression of the death receptor CD95 (Fig 1H). Collectively, these results show (1) that the increased frequency of advanced differentiation state CD8$^+$ T-cells in CMV+ individuals is not explained by the sheer presence of large, immunodominant CD8$^+$ CMV-specific T-cell populations, and (2) that part of the CMV-specific CD8$^+$ T-cells cluster separately from other CD8$^+$ T-cells.

## CMV infection affects expression of lifespan-associated markers in the CD4$^+$ T-cell pool

CD57 and KLRG-1 expression are often associated with decreased proliferative capacity [25,26] and CD95 (Fas) is known as a death receptor. Therefore, we assessed if, apart from the changes in T-cell numbers and phenotype, CMV infection also affected other lifespan-associated markers. T-cell proliferation was assessed by measuring the expression of Ki-67, and apoptosis resistance by measuring the expression of Bcl-2 (Fig 2A). Both CD4$^+$ T$_{EM}$ and T$_{EMRA}$ cells expressed significantly lower levels of Ki-67 in CMV+ compared to CMV- individuals (Fig 2A), suggesting that a lower percentage of cells were undergoing cell division in CMV+ individuals at the moment of blood withdrawal. Expression of the anti-apoptotic protein Bcl-2, although highly expressed in all populations, was significantly higher in CD4$^+$ T$_{EM}$ cells of CMV+ compared to CMV- individuals (Fig 2A). For CD8$^+$ T-cells, no significant differences in expression of these markers were observed between CMV+ and CMV- individuals (Fig 2A). As a proxy for lifespan, we combined the information of both previous flow cytometry panels (CD28, KLRG-1, CD57, Ki-67, and Bcl-2) in a principal component analysis (PCA), explaining 73.3% of the variance with two principal components. We found that CD4$^+$ T$_{EM}$ and T$_{EMRA}$ cells in CMV+ individuals clustered separately from CD4$^+$ T$_{EM}$ and T$_{EMRA}$ in CMV- individuals (Fig 2B). This was also the case for CD8$^+$ T$_{EM}$ cells, albeit to a lesser extent, but not for CD8$^+$ T$_{EMRA}$ cells (Fig 2C). Together, the effect of CMV infection on the expression of senescence and lifespan-associated markers was most pronounced for CD4$^+$ T$_{EM}$ and CD4$^+$ T$_{EMRA}$ cells, and suggestive of lower turnover of these cells in CMV+ individuals.

## CMV-specific CD8$^+$ T-cells most strongly resemble bulk CD8$^+$ T$_{EMRA}$ cells in lifespan-associated markers

We next investigated if CMV-specific CD8$^+$ T-cells differ in their expression of lifespan-associated markers compared to bulk memory CD8$^+$ T-cells. CMV-specific CD8$^+$ T$_{EM}$ cells expressed significantly less Ki-67 compared to bulk T$_{EM}$ cells ($P$ = 0.033) (Fig 2D). The

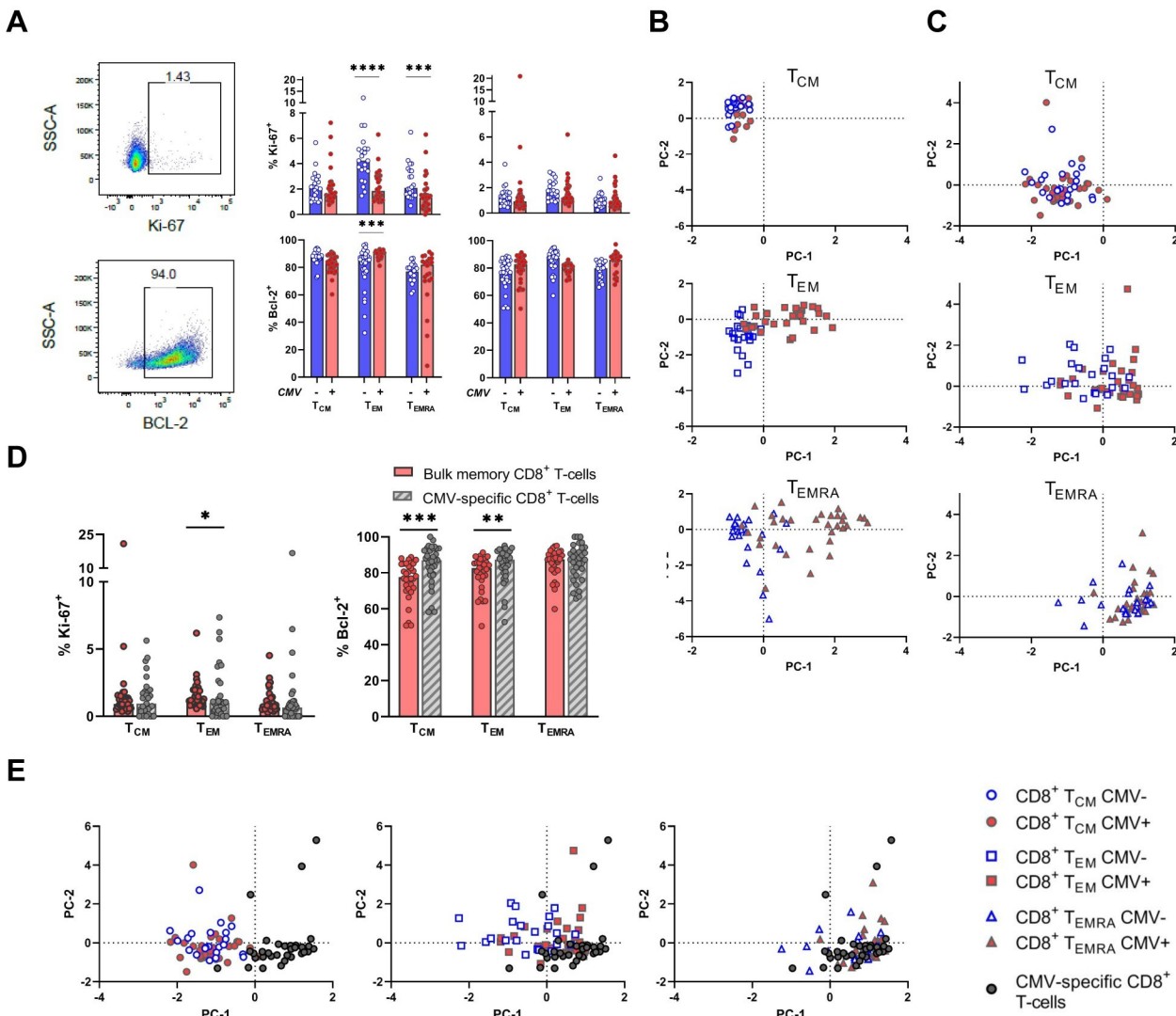

**Fig 2. Altered expression of lifespan-associated markers by CD4+ T_{EM/EMRA} cells in CMV infection. (A)** Representative flow cytometric plots for gating of Ki-67 and Bcl-2 expression. Graphs represent percentage of positive cells for Ki-67 and Bcl-2 in CMV- and CMV+ individuals by CD4+ and CD8+ T-cells, measured per memory subpopulation. **(B)** and **(C)** Principal component analysis (PCA) of CD4+ **(B)** and CD8+ **(C)** T-cells based on lifespan-associated markers KLRG-1, CD57, CD28, Ki-67, and Bcl-2 per memory subpopulation. Blue dots represent CMV- individuals, red dots represent CMV+ individuals. **(D)** Comparison of expression of Ki-67 and Bcl-2 by the total memory CD8+ T-cell pool of CMV+ individuals and by CMV-specific CD8+ T-cells, measured per memory subpopulation. **(E)** PC analysis of CD8+ T-cells based on lifespan-associated markers KLRG-1, CD57, CD28, Ki-67 and Bcl-2 per memory subpopulation. Bars in **(A)** and **(D)** represent medians. Differences between CMV- and CMV + individuals were tested by Mann-Whitney $U$ test. Stars indicate significant differences as follows: * $P$-value <0.05, ** $P$-value <0.01, *** $P$-value <0.001, **** $P$-value <0.0001.

expression of Bcl-2 was significantly higher for CMV-specific CD8+ T_{CM} and T_{EM} cells compared to bulk CD8+ T_{CM} and T_{EM} cells of CMV+ individuals ($P$ = 0.0003 and $P$ = 0.0092, respectively) (Fig 2D). Finally, we performed a PCA on these markers (CD28, KLRG-1, CD57, Ki-67, and Bcl-2) for total (dextramer+) CMV-specific CD8+ T-cells and the different memory CD8+ T-cell subsets (explaining 72.5% of variance with two principal components). We found that in terms of lifespan-associated markers, total CMV-specific CD8+ T-cells most strongly resemble bulk CD8+ T_{EMRA} cells and, to a lesser extent, bulk CD8+ T_{EM} cells (Fig 2E). In conclusion, it is not uncommon for CMV-specific CD8+ T-cells to express a T_{EM} or T_{CM}

phenotype, but in terms of lifespan-associated markers these cells most strongly resemble the phenotype of bulk $T_{EMRA}$ cells.

## Ten individuals selected for quantification of T-cell dynamics using heavy water labelling

The previously described markers only give a snapshot of apoptosis resistance and ongoing proliferation of different T-cell subsets in the peripheral blood. Therefore, to address the question whether circulating T-cells of CMV- and CMV+ individuals differ in their expected lifespans, and how immunodominant CMV-specific CD8$^+$ T-cell populations are maintained at such high cell numbers, we quantified the *in vivo* production and loss rates of different T-cell populations using heavy water ($^2H_2O$) labelling. The principle of $^2H_2O$ labelling is that the $^2$H-atoms present in the body are incorporated via *de novo* DNA synthesis by cells undergoing cell division and lost when cells die, differentiate, or migrate to another body compartment [27]. Ten individuals were selected to participate in the $^2H_2O$ labelling study (Fig 3A). We hypothesized that any potential differences in T-cell production and loss rates between CMV-specific CD8$^+$ T-cells and bulk memory CD8$^+$ T-cells in CMV+ and CMV- individuals would be most evident in individuals with the highest CMV-specific CD8$^+$ T-cell responses. We therefore selected the five CMV+ participants with the highest absolute numbers of CMV-specific CD8$^+$ T-cells based on dextramer-staining (S3 Fig). Next, five CMV- participants were selected based on matching for age and sex to the CMV+ group (Table 1). Percentages of memory and naive T-cells (S4 Fig) and expression of flow cytometric markers (S5 Fig) in these ten individuals were similar to those in the total group of participants.

In earlier deuterium labelling studies, we used quite broad definitions to separate memory (CD45RO$^+$) and naïve (CD45RO$^-$CD27$^+$) T-cell subsets. Here, we refined our sorting strategy to determine the production rates of different CD4$^+$ and CD8$^+$ T-cell subsets in all participants (S1 Fig). To analyze naive T-cells, we sorted truly naive ($T_{TN}$) cells (CD45RO$^-$CD27$^+$CCR7-$^{hi}$CD95$^-$) to exclude contamination with CD95$^+$ T-cells. CD95$^+$ T-cells are known to undergo quite frequent cell divisions [20] and include antigen-experienced stem cell memory T-cells ($T_{SCM}$) [28]. $T_{EM}$ and $T_{EMRA}$ cells were sorted together as $T_{EM/EMRA}$ to enable a fair comparison with CMV-specific T-cells.

Based on the longitudinal data of each sorted T-cell subset per individual during the 5 week up-labelling and on average 59 week (range 49–70) down-labelling phase of the $^2H_2O$ labelling study, the average production rate $p$ of each T-cell subset was estimated by fitting a mathematical model to the deuterium enrichment data (S6 and S7 Figs). For cell populations in steady state, $p$ represents both the (per capita) production rate and the average loss rate of cells. The estimated median $T_{TN}$ production/loss rates were 0.00065 per day (0.237 per year) for CD4$^+$ $T_{TN}$ cells and 0.00067 per day (0.245 per year) for CD8$^+$ $T_{TN}$ cells (Table 2 and S8 Fig). The average production/loss rates of $T_{CM}$ and $T_{EM/EMRA}$ cells were, respectively, 0.00740 and 0.00714 per day for CD4$^+$ T-cells (2.701 and 2.606 per year) and 0.00351 and 0.00279 per day for CD8$^+$ T-cells (1.281 and 1.018 per year), and thereby both for CD4$^+$ and CD8$^+$ T-cells higher than for their $T_{TN}$ counterparts (S8 Fig) as previously shown [20].

## The average production/loss rate of CD4$^+$ and CD8$^+$ memory T-cell subsets is comparable between CMV+ and CMV- individuals

Next, we investigated whether the increased number of memory T-cells and the upregulation of senescence markers seen in CMV infection were associated with altered T-cell dynamics. Therefore, the production/loss rates of $T_{CM}$ and $T_{EM/EMRA}$ CD4$^+$ and CD8$^+$ cells were compared between CMV- and CMV+ individuals. The deuterium enrichment curves of CMV+

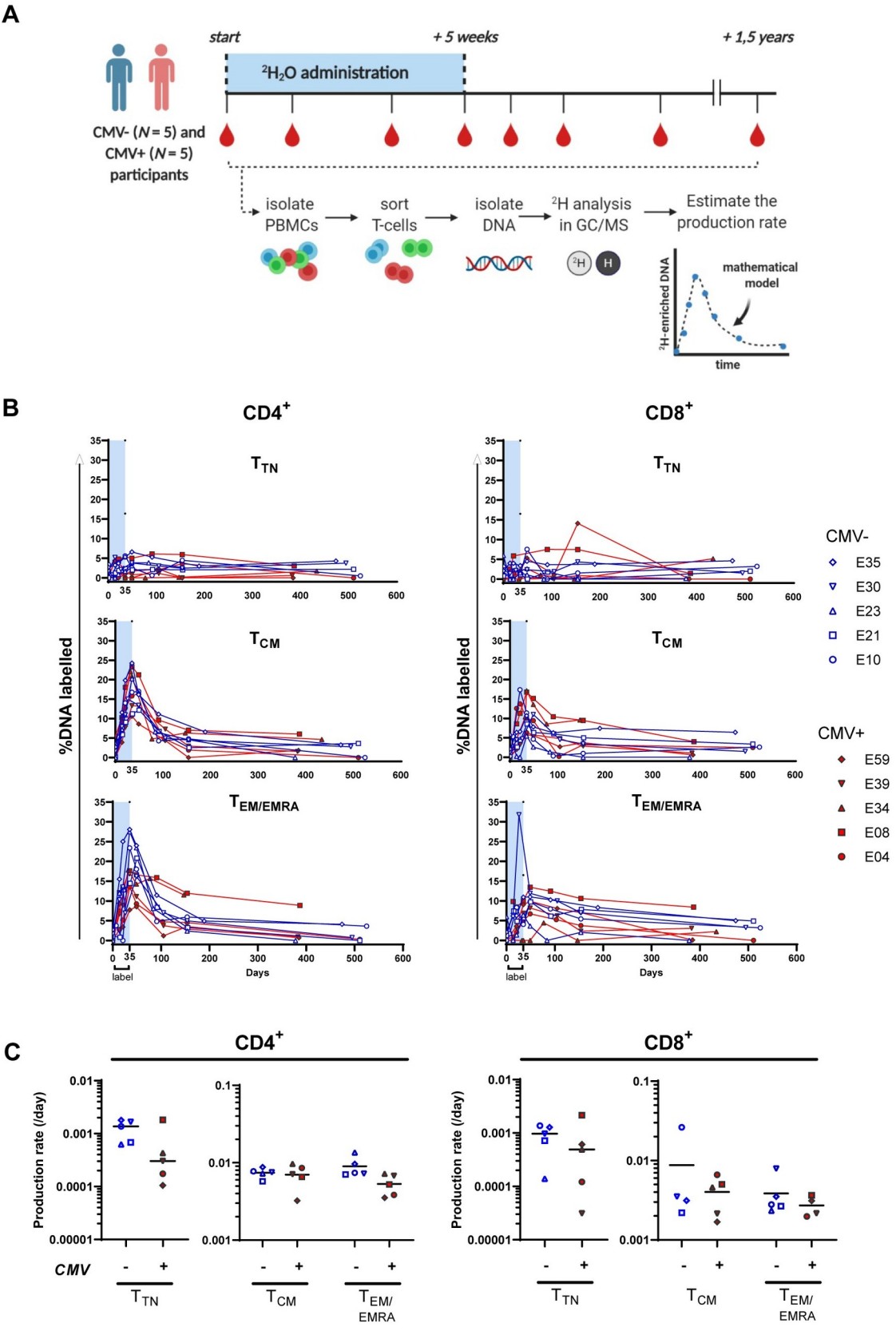

**Fig 3. Dynamics of CD4$^+$ and CD8$^+$ T-cells in CMV- and CMV+ individuals. (A)** Graphical representation of the study design for the deuterated water labelling protocol. Created with BioRender.com. **(B)** Deuterium labelling enrichment (%DNA labelled) of different T-cell subpopulations in CMV- and CMV+ individuals. Label enrichment was scaled between 0 and 100% by normalizing for the estimated maximum enrichment of granulocytes (see Materials and methods). Symbols depict the mean of duplicate measurements. **(C)** Summary of estimated production rates of $T_{TN}$, $T_{CM}$, and $T_{EM/EMRA}$ CD4$^+$ and CD8$^+$ T-cells in CMV- (blue symbols) and CMV+ (red symbols) individuals. All estimates were obtained by fitting a single-exponential model to the data sets per individual (see Materials and methods and S6 and S7 Figs). It was not possible to reliably estimate the production rate for $T_{CM}$ cells from E23 (CMV-) and $T_{EMRA}$ cells from E34 (CMV+). E30 only received $^2$H$_2$O for 3.5 weeks. Horizontal lines represent median values. Differences between groups were assessed by Kruskal-Wallis test. Data from each individual are represented by unique symbols.

and CMV- individuals were largely overlapping for CD4$^+$ $T_{TN}$ and $T_{CM}$ cells and for all CD8$^+$ T-cell subsets (Fig 3B). The enrichment curves for CD4$^+$ $T_{EM/EMRA}$ cells in CMV- individuals showed a steeper up- and down-labelling compared to those in CMV+ individuals (Fig 3B left lower panel). Although the difference in estimated production/loss rates of CD4$^+$ $T_{EM/EMRA}$ cells in CMV+ versus CMV- individuals did not reach statistical significance, the observed trend for a lower turnover rate of CD4$^+$ $T_{EM/EMRA}$ cells in CMV+ individuals (Fig 3C) is in line with our results based on Ki-67 and Bcl-2 expression (Fig 2A). In conclusion, the dynamics of most T-cell populations are largely unaltered by CMV infection.

## Kinetics of CMV-specific CD8$^+$ memory T-cells are similar to those of bulk memory T-cells

We investigated whether the high numbers of CMV-specific CD8$^+$ T-cells went hand in hand with altered production or loss rates. Therefore, we compared the dynamics of CMV-specific CD8$^+$ T-cells to those of $T_{EM/EMRA}$ CD8$^+$ T-cells. Strictly speaking, the value of $p$ derived from the deuterium enrichment curves of CMV-specific CD8$^+$ T-cells reflect their production rate. The production rate may not be equal to the loss rate if CMV-specific T-cell numbers increase during time (memory inflation) [13]. In our study, the percentage of CMV-specific T-cells in the total CD8$^+$ T-cell population remained fairly constant during the follow-up time of ~1.5 years (Fig 4A left panel). There were relatively large variations in absolute numbers of CMV-specific CD8$^+$ T-cells (Fig 4A right panel), which could be explained by similar variations in total leukocyte counts. Despite this, we observed no significant increase in the percentage or absolute number of CMV-specific CD8$^+$ T-cells over time. This implies that the average loss rate of CMV-specific CD8$^+$ T-cells should be close to their production rate $p$.

**Table 1. Selected individuals for heavy water labelling study.**

| | Sex | Age | CMV-specific IgG level | HLA type | | | | | | CMV-specific epitope |
|---|---|---|---|---|---|---|---|---|---|---|
| | | | | A1/A36 | A2 | A3 | A24 | B7 | B8 | |
| *Total CMV+* | | *69.14* | *781.58* | *2* | *3* | *2* | *2* | *2* | *1* | |
| E04 | Male | 68.8 | 185.9 | + | - | - | - | - | + | A1-VTE, B8-ELR |
| E08 | Male | 66.6 | 191.1 | + | + | - | + | - | - | A2-NLV |
| E34 | Male | 67.7 | 145.1 | - | + | - | - | - | - | A2-NLV |
| E39 | Male | 66.6 | 123.1 | - | - | + | - | + | - | B7-TPR |
| E59 | Female | 76 | 3262.7 | - | + | + | + | + | - | B7-TPR |
| *Total CMV-* | | *69.22* | *1.58* | *1* | *3* | *3* | *2* | *2* | *1* | |
| E35 | Female | 69.8 | 1.8 | - | + | - | - | - | - | N/A |
| E30 | Male | 66 | 1.4 | + | - | + | - | - | + | N/A |
| E23 | Male | 71.3 | 1.6 | - | - | + | + | + | - | N/A |
| E21 | Male | 69.7 | 1.5 | - | + | - | - | + | - | N/A |
| E10 | Female | 69.3 | 1.6 | - | + | + | + | - | - | N/A |

**Table 2. Median (range) of production rates ($p$) of the different T-cell subsets.**

| | Median (range) of production rates ($p$) per day | | |
|---|---|---|---|
| | *All individuals* | *CMV- individuals* | *CMV+ individuals* |
| **CD4$^+$ T$_{TN}$ cells** | 0.00065 (0.00011–0.00181) | 0.00137 (0.00063–0.00179) | 0.00030 (0.00011–0.00181) |
| **CD8$^+$ T$_{TN}$ cells** | 0.00067 (0.00003–0.00216) | 0.00097 (0.00014–0.00137) | 0.00049 (0.00003–0.00216) |
| **CD4$^+$ T$_{CM}$ cells** | 0.00740 (0.00322–0.00967) | 0.00755 (0.00578–0.00875) | 0.007098 (0.00322–0.00967) |
| **CD8$^+$ T$_{CM}$ cells** | 0.00351 (0.00168–0.02625) | 0.00332 (0.00220–0.02625) | 0.00460 (0.00168–0.00663) |
| **CD4$^+$ T$_{EM/EMRA}$ cells** | 0.00714 (0.00353–0.01359) | 0.00739 (0.00708–0.01359) | 0.00525 (0.00353–0.00720) |
| **CD8$^+$ T$_{EM/EMRA}$ cells** | 0.00279 (0.00198–0.00793) | 0.00279 (0.00234–0.00793) | 0.00263 (0.00198–0.00365) |
| **CMV-specific CD8$^+$ T-cells** | 0.00228 (0.00148–0.00659) | N/A | 0.00228 (0.00148–0.00659) |

For each individual, the incorporation of deuterium by CMV-specific CD8$^+$ T-cells was largely overlapping with that of total CD8$^+$ T$_{EM/EMRA}$ cells (Fig 4B upper panel). There is a risk that potential kinetic differences between CMV-specific CD8$^+$ T-cells and non-CMV-specific CD8$^+$ T-cells may go unnoticed in this comparison as bulk T$_{EM/EMRA}$ CD8$^+$ T-cells from CMV+ individuals in fact *include* many CMV-specific CD8$^+$ T-cells. We therefore also compared the dynamics of CMV-specific CD8$^+$ T-cells to those of bulk T$_{EM/EMRA}$ CD8$^+$ T-cells of CMV- individuals. Again, no clear differences in deuterium enrichment between CMV-specific CD8$^+$ T-cells and non-CMV-specific T$_{EM/EMRA}$ CD8$^+$ T-cells were observed (Fig 4B lower panel). Indeed, the estimated average production rates of CMV-specific CD8$^+$ T-cells were not significantly different from those of total CD8$^+$ T$_{EM/EMRA}$ cells in CMV+ or CMV- individuals (Fig 4C and Table 2). Although the sizes of the CMV-specific CD8$^+$ T-cell expansions differed more than threefold between individuals, we found no significant correlation between absolute numbers (or frequencies) of CMV-specific CD8$^+$ T-cells and their average production rates (Fig 4D). This suggests that the large size of clonally expanded CMV-specific CD8$^+$ T-cell subsets is neither due to a chronically increased T-cell production rate nor to a clearly decreased loss rate of these cells.

### Expression of Ki-67 and senescence markers are associated with *in vivo* production rates, while Bcl-2 expression is not

Finally, we studied if the estimated production rates of memory T-cell populations were associated with the expression of senescence, proliferation, and apoptosis markers. Interestingly, the expression of CD57, KLRG-1, and CD28 show moderate and highly significant correlations with the estimated production rates ($r = -0.6576$, $r = -0.6741$, $r = 0.7088$, respectively; for all markers $P < 0.0001$) (Fig 5). This suggests that advanced T-cell differentiation states are associated with reduced production rates *in vivo*. Expression of the proliferation marker Ki-67 correlated only weakly, but significantly with the *in vivo* production rate ($r = 0.4228$, $P = 0.0043$) (Fig 5). One individual (E04) showed surprisingly high Ki-67 expression levels given the corresponding production rates and seemed an outlier. Rerunning the analysis without this individual strengthened the correlation between Ki-67 expression and the estimated production rates ($r = 0.5718$, $P = 0.0002$). The expression of Bcl-2 was not significantly associated with the estimated production/loss rate. In conclusion, the expression of senescence markers and Ki-67 were indicative of T-cell production rates *in vivo*, although the example of E04 illustrates that these snapshot markers can be subject to large variations.

## Discussion

Immunological memory is indispensable for healthy ageing and it is therefore essential to understand the underlying biology of lymphocyte maintenance. CMV exerts a major effect on

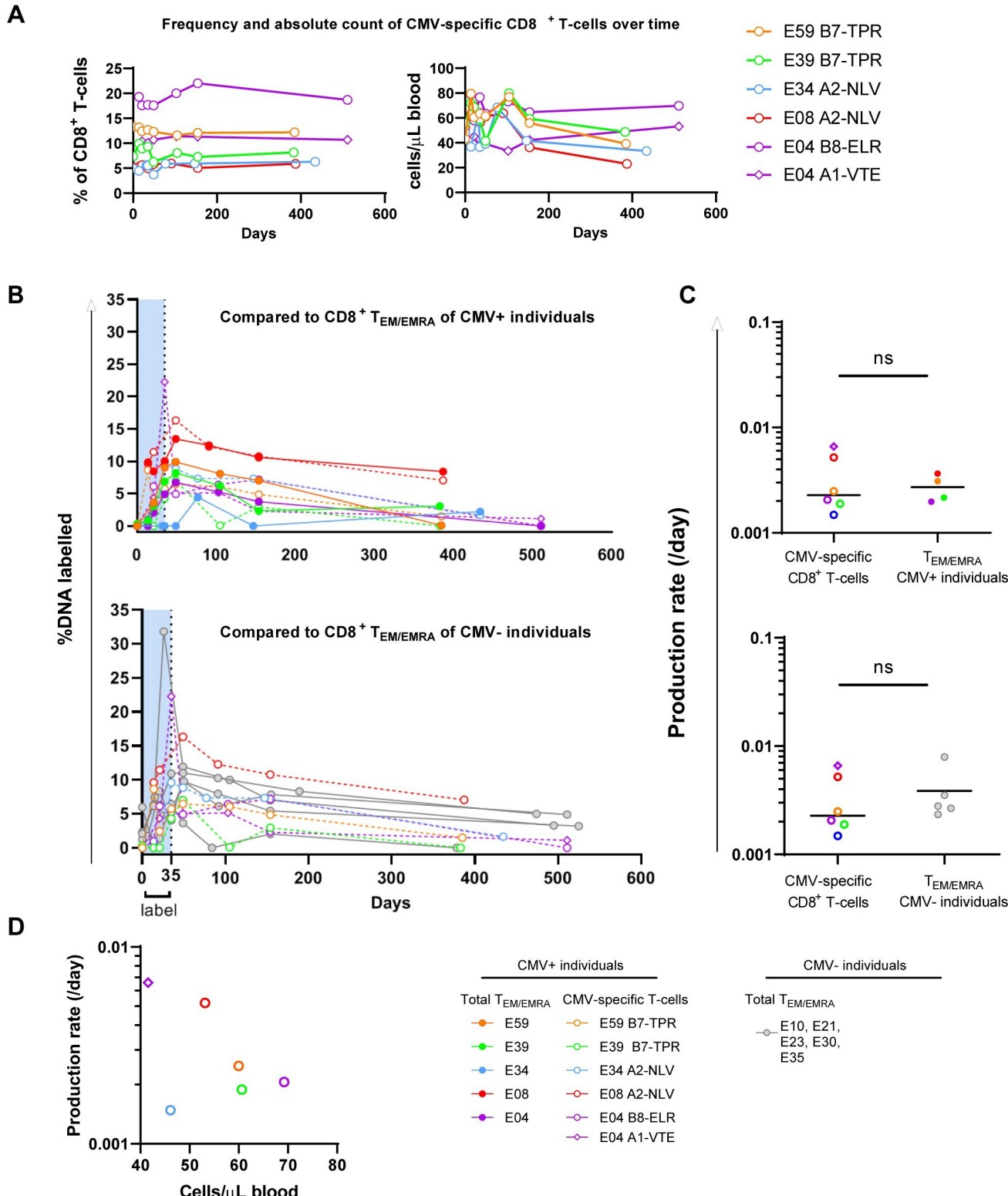

**Fig 4. Dynamics of CMV-specific CD8+ T-cells and CD8+ T_{EM/EMRA} cells in CMV+ and CMV- individuals. (A)** Frequency within the total CD8+ T-cell pool (upper panel) and absolute count (lower panel) of CMV-specific CD8+ T-cells over time in the CMV+ individuals selected for the deuterium labelling study. **(B)** Deuterium labelling enrichment (%DNA labelled) of CMV-specific CD8+ T-cells compared to CD8+ T_{EM/EMRA} cells of CMV+ (upper panel) or CMV- (lower panel) individuals. Label enrichment was scaled between 0 and 100% by normalizing for the estimated maximum enrichment of

granulocytes (see Materials and methods). Symbols depict the mean of duplicate measurements. It should be noted that for one individual (E34) it was not possible to determine the production rate of CD8+ T$_{EM/EMRA}$ cells because too many samples were out of range of the standard lines for deuterium enrichment measurement. E30 only received $^2H_2O$ for 3.5 weeks. **(C)** Estimated average production rates of CMV-specific CD8+ T-cells compared to CD8+ T$_{EM/EMRA}$ cells of CMV+ (upper panel) or CMV- (lower panel) individuals. All estimates were obtained by fitting a single-exponential model to the data sets per individual (see Materials and methods and S7 Fig). **(D)** Association between the absolute cell number and the average production rate of CMV-specific CD8+ T-cells. Horizontal lines represent median values. Differences between groups were assessed by Mann-Whitney *U* test, or Wilcoxon matched-pairs signed rank test for comparisons within the same individuals. Correlation was tested by Spearman's correlation. *P*-values < 0.1 are presented in the figure, ns = not significant. Data from each individual are represented by unique symbols.

the composition of the T-cell pool. The abundancy of antigen-specific CD8+ T-cells that are generated by CMV infection is unparalleled by any other antigenic stimulus, and CMV-specific T-cells display a unique late differentiated phenotype [9]. In this study, we investigated the effect of CMV infection on the dynamics of the T-cell pool in healthy older adults (Fig 6). Our key findings are that the CMV-induced late-stage differentiated state of the CD8+ T-cell memory pool is not explained by the sheer presence of large numbers of immunodominant CMV-specific CD8+ T-cells. In the CD4+ memory T-cell pool, latent CMV infection is associated with increased expression of late-stage differentiation markers and a marked decrease in expression of the proliferation marker Ki-67. Despite clear differences in the expression of late-stage differentiation markers, we found no significant differences in the *in vivo* production/loss rates of CD4+ and CD8+ memory T-cell subsets between CMV- and CMV+

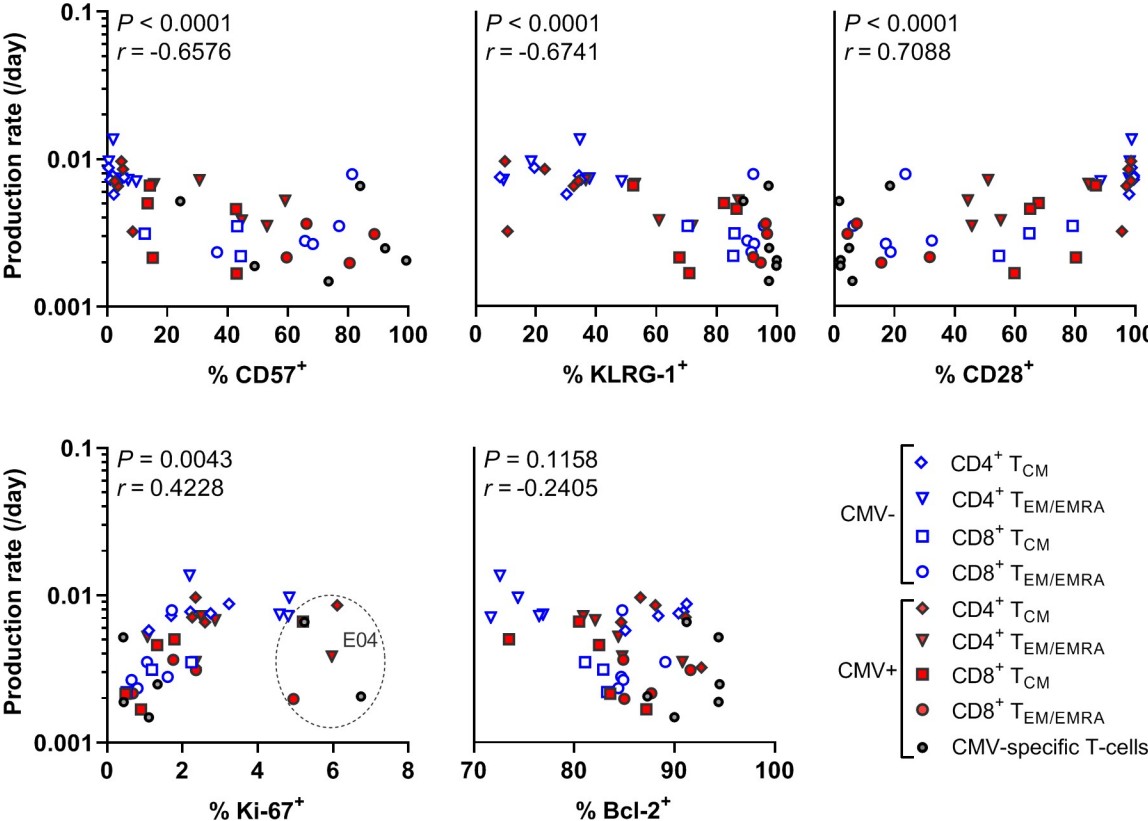

**Fig 5. Association between T-cell production rates and expression of senescence, proliferation, and apoptosis markers.** The estimated average production rates of different CD4+ and CD8+ T-cell subsets as estimated by heavy water labelling were plotted against their expression (% positive) of CD57, KLRG-1, CD28, Ki-67, and Bcl-2. Data points of donor E04 are circled in the Ki-67 graph. Correlation was tested by Spearman's correlation, *P*-values and *r*-values are indicated in the graphs. Different symbols represent different memory subpopulations.

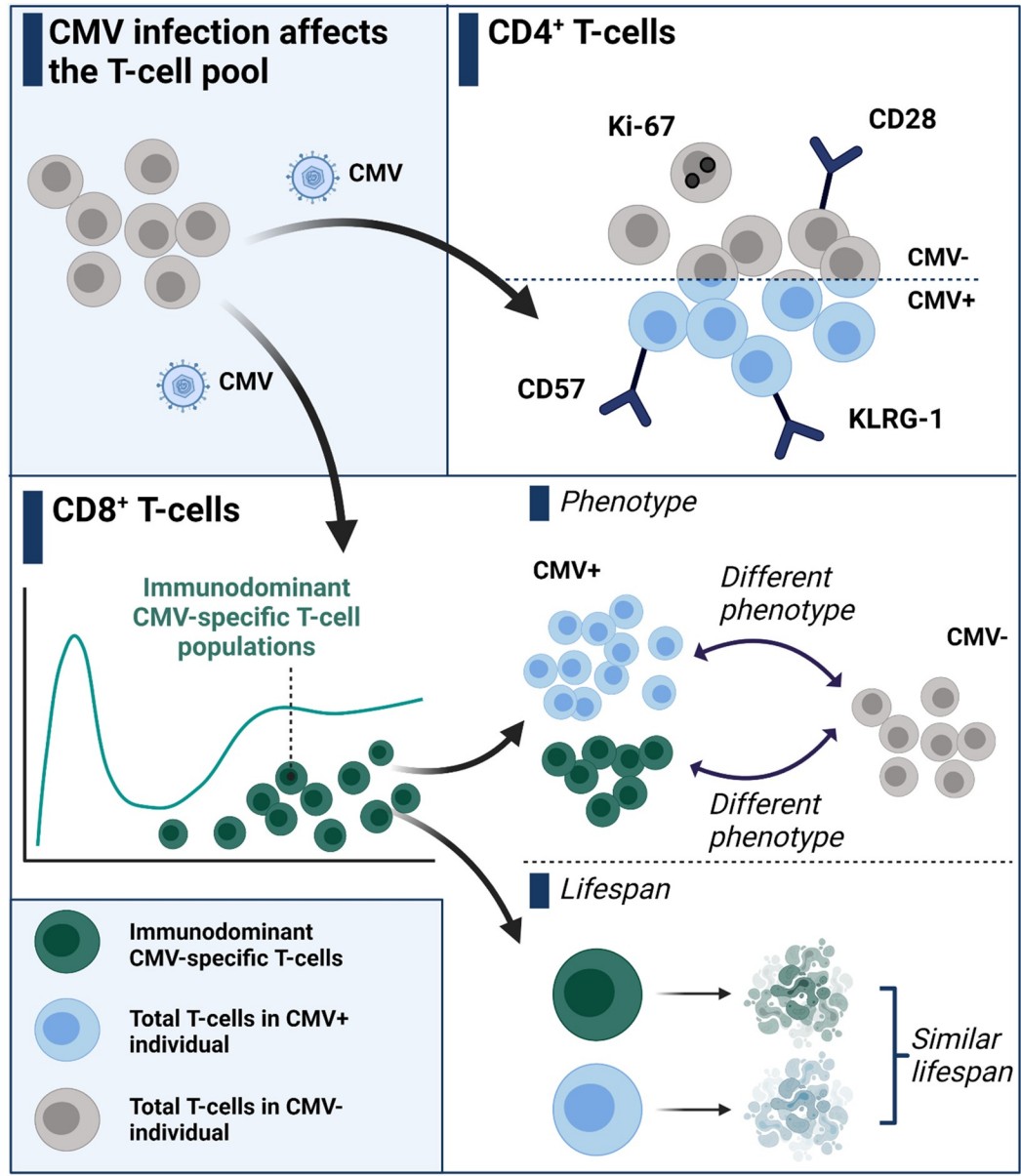

**Fig 6. CMV infection affects the phenotype but not the lifespan of T-cells in humans.** Both CD4+ and CD8+ T-cells in CMV-infected individuals upregulate the expression of senescence markers (e.g. CD57, KLRG-1) and downregulate the expression of co-stimulatory and proliferation markers (e.g. CD28, Ki-67). By separately analyzing the numerically large immunodominant CD8+ T-cell populations, we found that these do not explain the differences in T-cell phenotype between CMV+ and CMV- individuals. Though the increased late-differentiated state and size of the immunodominant CMV-specific CD8+ T-cell pool might suggest an altered lifespan, we find that these cells have similar lifespans as non-CMV-specific CD8+ T-cells. Created with BioRender.com.

individuals. Finally, despite their large number and late-stage differentiated phenotype, immunodominant CMV-specific CD8+ T-cell populations do not differ significantly in their *in vivo* production/loss rates when compared to $T_{EM/EMRA}$ or $T_{CM}$ cells in CMV- or CMV+ individuals.

Originally, the typical features of the T-cell pool in CMV+ individuals were thought to reflect an effect of CMV on the whole T-cell pool [24]. With the invention of MHC class I dextramers, however, it became clear that CMV-specific CD8$^+$ T-cells are present in exceptionally large numbers. As these cells *themselves* possess a phenotype that resembles the changes observed in the total T-cell pool of CMV+ individuals, we investigated whether their abundant presence could be the direct explanation for the phenotypical changes observed in CMV+ individuals. Based on t-SNE analysis, we found that the presence of large, immunodominant CMV-specific CD8$^+$ T-cell populations only partially explained the differences in the CD8$^+$ T-cell pool between CMV- and CMV+ individuals. Two different fluorochromes (PE and APC) for different epitopes of CMV-specific CD8$^+$ T-cells yielded comparable results in the t-SNE analysis, indicating that the mean fluorescence intensity (MFI)-based clustering of the CD8$^+$ T-cells was not biased by compensation for spectral overlap. This could mean that CMV also affects the phenotype of non-CMV-specific T-cells, as we have recently shown for EBV [29]. The investigated epitopes in our analysis captured at most 18% of CMV-specific CD8$^+$ T-cells out of the total CD8$^+$ T-cell pool, while some studies suggest that in CMV+ individuals as much as 30–90% of CD8$^+$ T-cells are CMV-specific [8,9]. We cannot formally exclude the possibility that CD8$^+$ T-cells specific for other, non-immunodominant CMV-epitopes may be responsible for the observed differences between CMV+ and CMV- individuals. We do not regard this very likely, however, because it would mean that non-immunodominant CMV-specific T-cells would have to have characteristics that differ from those of large, inflated CMV-specific T-cell populations but also from those of T-cells with other specificities. Ideally, a method that would pick up all CMV-specific CD8$^+$ T-cells would need to be used to reveal the precise contribution of CMV-specific T-cells to the differences in the T-cell pool observed between CMV+ and CMV- individuals, but this is technically very challenging.

Even though most research has focused on the large numerical expansions of CMV-specific CD8$^+$ T-cells, also the CD4$^+$ T-cell pool differs significantly between CMV+ and CMV- individuals. As we did not use peptide-MHC class II dextramers, we could not quantify to what extent CMV-specific CD4$^+$ T-cells were responsible for these differences and to what extent CMV affected non-CMV-specific memory CD4$^+$ T-cells. We found that the expression of late-stage differentiation markers by CD4$^+$ T-cells was higher in CMV+ individuals, whereas their expression of the proliferation marker Ki-67 was markedly decreased. Although the estimated production/loss rates based on in vivo deuterium labeling were not significantly different between CMV+ and CMV- individuals, we observed considerably lower deuterium incorporation in CMV+ individuals. Together, these observations point towards a lower turnover rate for the late-differentiated CD4$^+$ T$_{EM/EMRA}$ cells in CMV+ individuals. It is tempting to speculate that the CD4$^+$ T$_{EM/EMRA}$ cell population in CMV+ individuals mainly consists of CMV-specific cytotoxic T-cells. Cytotoxic CD4$^+$ T-cells express a T$_{EM/EMRA}$ phenotype and their numbers are typically increased in CMV+ compared to CMV- individuals [30,31]. In line with this, CD8$^+$ T-cells, which are known to have a larger cytotoxic potential than CD4$^+$ memory T-cells, have a longer expected lifespan than CD4$^+$ T-cells [20], suggesting that cytotoxic function itself may be related to longevity.

It was previously proposed that chronic infections may lead to increased T-cell production rates [32]. Our previous observation that the only CMV+ individual in a deuterated water labelling study [20] had higher CD8$^+$ memory T-cell production rates than the other four CMV- individuals in the study seemed to point in the same direction. Alternatively, it has been proposed that CMV-specific CD8$^+$ T-cells might outcompete other pre-existing antigen-specific T-cells [33–35], and thereby affect their lifespan. Despite the increased CD8$^+$ T-cell numbers and their late-stage differentiation phenotype in CMV+ individuals, we found that neither the expression of CD8$^+$ T-cell proliferation and apoptosis markers, nor the *in vivo*

production/loss rates of memory CD8$^+$ T-cells as determined by deuterium labelling differed significantly between CMV+ and CMV- individuals. Thanks to our rather strict health selection criteria, we were able to study the influence of CMV infection on the T-cell pool in the absence of other underlying (chronic) conditions. On the other hand, we do not know the extent of latency-associated viral gene expression [36], and whether partial viral reactivation [37,38] occurred during follow-up in our participants. We cannot exclude the possibility that in these extremely healthy individuals partial CMV reactivation may have occurred less frequently than in individuals with impaired health, which may in turn influence the dynamics of the CD8$^+$ T-cell pool.

We also assessed if the high numbers of CMV-specific CD8$^+$ T-cells can be explained by increased production or decreased loss rates of these cells. In the deuterated glucose labelling study by Wallace *et al.*, the dynamics of tetramer+ CMV-specific T-cells were compared to those of tetramer-negative T-cell subsets (CD45RO$^+$ (i.e. T$_{EM}$ and T$_{CM}$) and CD45RO$^-$ (i.e. T$_N$ and T$_{EMRA}$)) within the same individual. The labelling curves of tetramer+ CMV-specific T-cells were found to be lower than those of CD45RO$^+$ and CD45RO$^-$ T-cells and it was concluded that CMV-specific T-cells are longer-lived than CD45RO$^+$ and CD45RO$^-$ T-cells [19]. In contrast, we found enrichment curves that are largely overlapping between CMV-specific and bulk memory CD8$^+$ T-cells. A clear difference between both studies is that we estimated the production rates of more strictly defined naive and memory T-cell populations to exclude mixing of kinetically heterogeneous subsets, e.g. T$_N$ and T$_{EMRA}$ cells. The contrast between the results of these two studies remains puzzling for two reasons. First, our approach should be able to pick up smaller differences in production rates of long-lived cells because we used long-term $^2$H$_2$O labelling, instead of short-term deuterated glucose labelling [39]. Second, we compared the production rates of CMV-specific CD8$^+$ T-cells not only to bulk CD8$^+$ memory T-cell subsets from CMV+ individuals but also from CMV- individuals. The latter ensures a comparison to a memory T-cell pool that is definitely devoid of CMV-specific T-cells, whereas this is not true for the former. Nevertheless, we did not find significant differences in the production rates of CMV-specific CD8$^+$ T-cells in comparison with CD8$^+$ T$_{EM/EMRA}$ or T$_{CM}$ cells from CMV+ or CMV- individuals. In line with this, in a recent study of murine CMV, we also find that the *in vivo* production rates of MCMV-specific CD8$^+$ T-cells were very similar to those of bulk memory CD8$^+$ T-cells [40].

What do these similar labelling curves of CMV-specific and bulk memory T-cells tell us? Firstly, since CMV-specific T-cell clones tend to be of much larger size than other memory clones, the *total number* of CMV-specific cells that is produced per day is expected to outnumber that of other memory specificities. Secondly, assuming that the vast majority of CMV-specific T-cells are formed by peripheral T-cell proliferation, and not by continuous recruitment of naive CMV-specific T-cells into the memory pool, these findings imply that CMV-specific T-cells do not divide more frequently than other memory T-cells, which is supported by their Ki-67-expression levels. Thirdly, assuming that CMV-specific T-cell numbers were indeed no longer increasing over time in these individuals, the overlap in deuterium labelling curves implies that the expected lifespans of CMV-specific and bulk memory T-cells are similar. We thus find no evidence for the previously proposed idea that CMV-specific T-cells are longer-lived than other memory T-cells [19]. The latter claim is somewhat harder to make because it depends on the measurement of cell numbers, which are notoriously variable, and we cannot rule out the possibility that our follow-up time of 1.5 years may have been too short to reliably measure a small increase in cell numbers over time. Yet, our data suggest that there are no large differences in the expected lifespans of CMV-specific CD8$^+$ T-cells and bulk memory CD8$^+$ T-cells. Although we cannot rule out the possibility that small differences in production/loss rates may have been missed in our *in vivo* deuterium labelling study, our data suggest that

high CD8$^+$ CMV-specific T-cell numbers do not go hand in hand with substantial differences in the average production or loss rates of these cells. The large size of the CMV-specific T-cell pool is possibly set shortly after CMV infection [41], though the underlying mechanisms explaining these large CMV-specific expansions remain to be elucidated.

Deuterium labeling studies tend to be limited to relatively small numbers of participants, which makes it more difficult to draw firm conclusions on kinetic comparisons between groups with a lot of inter-individual variation. Nevertheless, deuterium labelling provides a powerful tool to quantify cellular dynamics, not in the least because kinetic estimates are based on multiple data points per individual, making the estimates per individual less sensitive to fluctuations over time. As a result, deuterium labelling studies, even when based on relatively small numbers of participants, can yield very reliable and reproducible results. We and others have for example reproducibly shown that CD4$^+$ T-cells have higher production rates than CD8$^+$ T-cells [42]. The kinetic heterogeneity model that we used to fit the data [43] has the disadvantage that the estimated average production rate can be somewhat dependent on the length of the labelling period [39]. We nevertheless used it, because the multi-exponential model that we previously proposed [44] frequently led to overfitting of the data. This model choice may have affected the estimated production rates of participant E30 who -unlike all other participants- only received 3.5 weeks of heavy water labelling. Importantly, when using the multi-exponential model, we also did not find any significant differences in the production rates of CD8$^+$ CMV-specific T-cells and bulk CD8$^+$ T$_{EM/EMRA}$ cells (S8E Fig). Our results thus seem independent of the exact model choice. A downside of deuterium labelling is that it cannot be used to study single-cell kinetics and needs to be performed on sorted populations based on predefined cell surface markers. It could be that kinetically different subpopulations cannot be subdivided along the same lines as the well-established memory T-cell populations (e.g. T$_{CM}$, T$_{EM}$, T$_{EMRA}$), and that other markers or functionalities govern cellular production and lifespan.

Finally, we put the investigated *in vivo* production and loss rates in context of the expression of proliferation, apoptosis, and senescence markers. We found that the estimated *in vivo* production rates of memory T-cells based on deuterium labelling correlated with the expression of late-stage differentiation markers and the proliferation marker Ki-67. Expression of Ki-67 correlated weakly with the *in vivo* production rate, possibly due to the 'snapshot' nature of this marker, which makes it more sensitive to day-to-day variations and intermittent immune activations. The expression of the late-stage differentiation markers CD57 and KLRG-1 correlated negatively with the *in vivo* production rates. This is in line with *in vitro* results, which showed that CD57$^+$ and KLRG-1$^+$ T-cells are less capable of proliferating upon *in vitro* T-cell receptor (TCR) stimulation [25,26]. However, in mice, it was shown that CD57$^+$ as well as CD57$^-$ antigen-specific T-cells are able to divide upon IL-7 stimulation, a measure for homeostatic proliferation, and are therefore not hampered in their maintenance [45]. Hence, it would be interesting to directly study the *in vivo* production rate of sorted CD57$^+$ and CD57$^-$ cells. Altogether, our data suggests that a combination of senescence markers could potentially be used as a proxy marker for cellular production and loss rates.

In the context of ageing of the immune system, or immuno-senescence, potential differences in T-cell maintenance between CMV- and CMV+ individuals are of special interest. CMV has been suggested to be a driving force behind accelerated ageing of the immune system [46–48]. Previously, we did not find a difference in memory T-cell maintenance between aged and young individuals [20]. Here we expand on this finding by showing that infection with CMV at most has minor effects on T-cell dynamics. Together, this suggests that during healthy ageing, with or without CMV infection, no substantial changes in memory T-cell dynamics arise. Secondly, our work shows that maintenance of CMV-specific CD8$^+$ T-cells, despite their

high abundance, does not go hand in hand with altered cellular dynamics. This suggests that the characteristics of large memory T-cell responses may be set at the start of the T-cell response. This knowledge could contribute to the development of CMV-vector based vaccines, as the long-term maintenance of CMV-specific CD8$^+$ T-cells is key for successful CMV-vector based vaccination strategies. Once a large CMV-specific T-cell pool has been established, it is likely that it can persist much like any other memory T-cell population.

## Materials and methods

### Ethics statement

This study was approved by the Medical Ethical Committee of the University Medical Center Utrecht (UMCU) (study approval number 15/745), The Netherlands, and was conducted in accordance with the Helsinki Declaration, last amended in 2013. All participants gave written informed consent.

### Study design

Fifty-four healthy older adults were included based on very strict health criteria to ensure that we were investigating a steady-state situation. Individuals were excluded from participation if they had a condition that could influence the immune system (e.g. infection with human immunodeficiency virus, hepatitis B or C virus, Lyme disease, malaria, or chronic diseases such as asthma or diabetes mellitus, or (a history of) cancer) or drug use (with the exception of occasional use of paracetamol or ibuprofen)). For the heavy water labelling study, we selected five CMV+ individuals with high dextramer-positive T-cell responses against CMV, as well as five age-matched CMV- individuals. We obtained heparinised blood by venepuncture from all fifty-four participants at one time point. The ten participants that were included in the deuterium labelling study donated blood an additional eight times.

### Cytomegalovirus (CMV)-specific antibodies

CMV-specific antibody (Ab) levels were measured in serum by a sensitive multiplex immunoassay [49]. A cut-off value of 5 (RU) mL$^{-1}$ was used to define CMV-seropositivity. To decrease the chance of false-positive or false-negative results, we only considered samples to be CMV-seronegative if the Ab level was $\leq 4$ (RU) mL$^{-1}$ and CMV-seropositive if the Ab level was >7.5 (RU) mL$^{-1}$. All of our samples were clearly CMV-seropositive or CMV-seronegative according to these definitions.

### Cell numbers and HLA typing by flow cytometry

Whole blood was used to calculate absolute leukocyte counts by Trucount analysis (BD Biosciences) according to manufacturer's protocol. The following antibodies were used: CD45-PerCP (BioLegend), and CD3-APC-R700, CD4-BV711, CD8-BV786, all purchased from BD Biosciences. Using the bead count of the Trucount tube and the CD3$^+$ cell count, absolute cell numbers of the proliferation/apoptosis and senescence panel were also calculated. In addition, for all individuals, HLA typing on whole blood was performed by flow cytometry at the first visit. For all these HLA-types, we characterized the known immunodominant CMV epitopes for which commercial dextramers are available. Antibodies used were HLA-A1/ 36-biotin/strep-PE-CF594, HLA-A2-V450, HLA-A3-APC, HLA-A24-PE, HLA-B7-FITC, and HLA-B8-PE-Cy7. Cells were measured on a Fortessa flow cytometer (BD Biosciences) and analyzed with FlowJo V10 software.

## PBMC, neutrophilic granulocytes, and serum isolation and storage

Peripheral blood mononuclear cells (PBMCs) were isolated using Ficoll-Paque PLUS (GE) density centrifugation according to the manufacturer's protocol. After isolation of the PBMCs, cells were washed with phosphate buffered saline (PBS) supplemented with 0.2% FCS, and then frozen in a solution with 90% fetal calf serum and 10% dimethyl sulfoxide in liquid nitrogen or -135˚C until further use. Neutrophilic granulocytes were obtained from the Ficoll-Paque pellet after erythrolysis with shock buffer (155 mM $NH_4Cl$, 10 mM $KHCO_3$, and 0,1 mM EDTA in Aqua Millipore). Serum was collected separately and stored at -80˚C.

## Proliferation/Apoptosis and senescence markers by flow cytometry

Thawed PBMCs of all 54 individuals were stained with a proliferation/apoptosis and a senescence panel for flow cytometry. The apoptosis panel consisted of an extracellular antibody mix (CD3-PerCP and CD45RO-PE-Cy7 (both from Biolegend), CD27-APC-R700, Live/dead-APC-Cy7, CD28-BV510, CD56-BV711, CD8-BV786, CCR7-BUV395, CD4-BUV737 (all from BD Biosciences)), and after fixing and permeabilising (Cytofix/Cytoperm; BD Biosciences) an additional intracellular antibody mix (Ki-67-FITC and Bcl-2-PE/Dazzle594 (both from BD Biosciences)). Washing steps were performed using Perm/Wash buffer (BD Biosciences). The senescence panel contained CD57-FITC (eBioscience), CD3-AF700, Live/dead-APC-Cy7, CD95-BV421, CD8-BV510, CD27-BV786, CCR7-BUV395, and CD4-BUV737 (all from BD Biosciences), CD127-BV650, CD45RO-BV711, and KLRG1-PE-Cy7 (all from Biolegend). In all flow cytometric panels, CMV+ donors were first stained with the specific dextramer, either with HLA-B*0702/TPRVTGGGAM-PE, or HLA-A*0201/NLVPMVATV-PE or -APC, or both HLA-A*0101/VTEHDTLLY-APC and HLA-B*0801/ELRRKMMYM-PE (all from Immudex), for 30 minutes at room temperature. An overview of CMV-dextramers used per donor can be found in S1 Table. Cells were analyzed on a Fortessa flow cytometer (BD Biosciences) and data analysis was performed with FlowJo V10 software. T-SNE analysis was performed on 10.000 CD8+ T-cells from every donor using the senescence panel for CD8+ T-cells (cytobank.org).

## *In vivo* deuterated water labelling protocol

Ten participants (five CMV+ and five CMV-) were included in a longitudinal heavy water labelling study to quantify the underlying dynamics, i.e. production and loss rates, of different sorted T-cell subsets. In short, $^2H$ is incorporated via *de novo* DNA synthesis by cells undergoing cell division and lost when cells die, differentiate, or migrate to a different body compartment [27]. Participants received a ramp-up dose of $^2H_2O$ on the first day, after which they drank a daily dose of $^2H_2O$ for five weeks (except for participant E30 who drank $^2H_2O$ for only three and a half weeks for logistical reasons). The procedures on the first day and the follow-up schedule are similar to those described before [20], with the exception that the length of labelling was reduced to five (rather than nine) weeks to reduce the burden for the participants. Urine was collected at thirteen time points during the up- and down-labelling phases to correct for $^2H_2O$-availability in the body. Blood was collected to sort T-cell subsets at eight time points, of which four during up-labelling and four during down-labelling, with an average follow-up time of 448 days (range 378 to 511 days).

## Sorting of T-cell subpopulations by flow cytometry

Directly after Ficoll-Paque isolation, between $5 \cdot 10^7$ and $20 \cdot 10^7$ PBMCs were stained with CD95-FITC, CD4-APC-eF780 (both eBioscience), CD3-PerCP, CCR7-BV421, CD8a-BV510,

and CD45RO-PE-Cy7 (all from BioLegend), and either CD56-APC, CD56-PE, or CD56-PE/Dazzle-594 (the first purchased from BD Biosciences and the latter two from BioLegend). T-cell subpopulations were defined as follows: truly naive, $T_{TN}$ (CCR7$^+$CD45RO$^-$CD27$^+$CD95$^+$), central memory, $T_{CM}$ (CD45RO$^+$CD27$^+$), effector memory, $T_{EM}$ (CD27$^-$CD45RO$^+$) and effector memory re-expressing RA, $T_{EMRA}$ (CD27$^-$CD45RO$^-$). The full gating strategy can be found in S1 Fig. Cells were sorted on a FACSAria II or FACSAria III sorter (BD Biosciences), and data was analyzed with the BD FACSDiva v8.0.1 and FlowJo V10 software.

## DNA isolation and measurement of deuterium enrichment by GC/MS

After sorting of the different T-cell populations, DNA was isolated after overnight storage of the cell pellets at 4˚C using the ReliaPrep DNA isolation kit (Promega). DNA samples were stored in 200 μL DNAse free water at -20˚C until further processing. Next, we followed the protocols described previously [20,27], with minor modifications, to obtain the percentage deuterium enrichment in body water and immune cell populations. Briefly, DNA samples from sorted lymphocytes were enzymatically hydrolyzed into deoxyribonucleotides and conjugated to pentafluorotriacetate (PFTA). The PFTA derivative of deoxyadenosine was analyzed by measuring the ions m/z 435 u (M+0) and m/z 436 u (M+1) in the gas chromatograph-mass spectrometer (GC/MS) (7890A GC System, 5975C inert XL EI/CI MSD with Triple-Axis Detector; Agilent Technologies). The tracer-to-tracee ratio (TTR) was then calculated, by dividing the enriched, deuterated ion (M+1) by the unenriched, naturally occurring ion (M+0).

## Correction for abundance sensitivity of the GC/MS

The measured TTR of each sample was corrected for abundance sensitivity, that is the positive correlation observed between sample input and *measured* TTR at a fixed *theoretical* TTR [27,50]. In short, eight standards with different known enrichments ([M+1]/[M+0] = 0, 0.0016, 0.0031, 0.0063, 0.0126, 0.0255, 0.0523, and 0.01097) were measured on the GC/MS at different sample input (M+0), within the same time period as the samples (<3 months). A second-order polynomial was fitted to each standard (M+0 versus M+1), and the corrected TTR of each sample was calculated by linear interpolation between these polynomials [50]. All data were arcsin(sqrt) transformed before fitting different models to the data (see below). To allow for this transformation, any negative values had to be set to zero. Finally, the atom percent excess (APE) of each sample was calculated from the corrected TTR values (as APE = TTR/(1+TTR)) and presented in the paper as the fraction enriched DNA. Data points that were out of the M+0 range of the standards ($N$ = 317/1059) were excluded from the analysis and are indicated in red in S6 and S7 Figs.

## Mathematical modelling to estimate T-cell production and loss rates

We first fitted a simple label enrichment/decay curve to the urine enrichment data of each individual, during label intake ($t \leq \tau$) (Eq 1) and after label intake ($t > \tau$) (Eq 2):

$$U(t) = f(1 - e^{-\delta t}) + \beta e^{-\delta t} \tag{1}$$

$$U(t) = [f(1 - e^{-\delta \tau}) + \beta e^{-\delta \tau}]e^{-\delta(t-\tau)} \tag{2}$$

as described previously [20], where *U(t)* represents the fraction of $^2$H$_2$O in urine at time *t* (in days), *f* is the fraction of $^2$H$_2$O in the drinking water, labelling was stopped at *t* = $\tau$ days, $\delta$ represents the turnover rate of body water per day, and $\beta$ is the plasma enrichment due to the

boost of label at day 0 (S9A and S9B Fig). We used the parameters of the best fits to these data when analyzing the enrichment in the different cell populations. The estimated maximum level of $^2$H enrichment in the granulocyte population of each individual was considered to be the maximum level of label incorporation that cells can possibly attain, and was used to scale the enrichment data of the other cell subsets. Because the granulocyte enrichment curves had not yet reached plateau after 5 weeks of labelling (S9C Fig), we constrained the fits to the granulocyte data by imposing that all individuals have the same granulocyte turnover rate $d$ and a fixed delay of bone marrow maturation ($\Delta$) of 5 days [51] in the following differential equation, describing the granulocyte dynamics for all individuals:

$$\frac{dL}{dt} = dcU(t - \Delta) - dL \tag{3}$$

where $L$ represents the fraction of labelled DNA in the cell population, here the granulocyte population. All T-cell deuterium labelling curves were fitted using the kinetic heterogeneity model [43]:

$$\frac{dL}{dt} = pcU(t) - dL \tag{4}$$

where $L$ again represents the fraction of labelled DNA, here in the T-cell population of interest, yielding the average per capita production rate $p$ of the T-cell population, and the average loss rate of the *labelled* T-cells $d^*$ for each cell subset (S2 Table). For cell populations in steady state, $p$ represents both the (per capita) production rate and the loss rate, which can be translated into the "average lifespan" of the cells by calculating $1/p$. The estimates of $p$ are presented in this paper either as rates per day (in tables and figures), or as rates per year (by multiplying $p$ with 365).

## Statistical analysis

Differences between groups were assessed using a Mann-Whitney $U$ or Kruskal-Wallis test, and comparisons within the same individuals with the Wilcoxon singed-rank test. Correlations were tested with Spearman's rank correlation coefficient. Principal component analysis (PCA) was performed in SPSS, reducing 5 variables to less factors of an eigenvalue >1, iteration was set at a maximum of 25. For all analyses $P$-values <0.05 were considered significant. Data were analyzed using GraphPad Prism 8.3 and SPSS statistics 22 for Windows (SPSS Inc., Chicago, IL, USA). Deuterium-enrichment data were fitted using R V3.6.1, parameters were estimated using a maximum likelihood approach. Using a bootstrap method, the 95% confidence limits of the parameter were determined by resampling 500 times the residuals to the optimal fit.

## Supporting information

**S1 Fig. Gating strategy for sorting of different T-cell populations.** Peripheral blood mononuclear cells (PBMCs) were isolated using Ficoll-Paque PLUS density centrifugation and stained for CD3, CD56, CD4, CD8, CD45RO, CCR7, CD27, and CD95 (see Materials and methods for antibody specificities). Live lymphocytes were gated based on SSC-A/FSC-A and FSC-H/FSC-A plots. T-cells were subsequently defined as CD3$^+$CD56$^-$. T-cell subpopulations were defined as follows: truly naive, $T_{TN}$ (CCR7$^+$CD45RO$^-$CD27$^+$CD95$^+$), central memory, $T_{CM}$ (CD45RO$^+$CD27$^+$), effector memory, $T_{EM}$ (CD27$^-$CD45RO$^+$) and effector memory re-expressing RA, $T_{EMRA}$ (CD27$^-$CD45RO$^-$).
(TIF)

**S2 Fig. Percentage of CD8$^+$ T-cells per cluster in the t-SNE analysis.** CD8$^+$ T-cells of CMV- and CMV+ individuals were clustered by t-SNE analysis based on the expression of CD57, KLRG-1, CD127, CD27, CD45RO, CCR7, and CD95 (Fig 1E). Nine clusters were identified within the t-SNE plot based on cell density. **(A)** The percentage of CD8$^+$ T-cells out of total CD8$^+$ T-cells that fall into each cluster is shown for CMV- and CMV+ individuals. **(B)** The percentage of CMV-specific CD8$^+$ T-cells out of total CMV-specific CD8$^+$ T-cells that fall into each cluster is shown. We only included data points when more than 20 cells could be analyzed. Differences between CMV- and CMV+ individuals were tested by Kruskal-Wallis test. Stars indicate significant differences as follows: * $P$-value <0.05, ** $P$-value <0.01, *** $P$-value <0.001, *** $P$-value <0.0001. (TIF)

**S3 Fig. Number and frequency of CMV-specific T-cells.** Absolute numbers (in cells/μL) and frequency (in % of total CD8$^+$ T-cells) of CMV-specific CD8$^+$ T-cells are shown for all 32 CMV+ individuals. For the heavy water labelling study, five CMV+ participants were selected based on highest absolute CMV-specific T-cell numbers (shown with individual symbol). CMV+ participants that were not selected are shown in grey. (TIF)

**S4 Fig. Percentages of T-cells in all participants and in the ten selected for the heavy water labelling study. (A)** and **(B)** The frequency (left panels) and absolute numbers (right panels) of different CD4$^+$ **(A)** or CD8$^+$ **(B)** T-cell populations. In the top panels of both **(A)** and **(B)** all participants are shown, and in the lower panels only those selected for the heavy water labelling study, highlighted with a unique color. Differences between CMV- and CMV+ individuals were tested by Mann-Whitney $U$ test. Stars indicate significant differences as follows: * $P$-value <0.05, ** $P$-value <0.01, *** $P$-value <0.001, *** $P$-value <0.0001. (TIF)

**S5 Fig. Percentages of senescence, proliferation, and apoptosis markers in all participants and the ten selected for the heavy water labelling study. (A)** and **(B)** The percentage of T$_{CM}$ and T$_{EM/EMRA}$ positive for CD57, KLRG-1, CD28, Ki-67, and Bcl-2 are shown for CD4$^+$ **(A)** or CD8$^+$ **(B)** T-cell populations. In the top panels of both **(A)** and **(B)** all participants are shown, and in the lower panels only those selected for the heavy water labelling study, highlighted with a unique color. **(C)** The expression of CD57, KLRG-1, CD28, Ki-67, and Bcl-2 on CMV-specific CD8$^+$ T-cells is compared to the expression on bulk T$_{EM/EMRA}$ CD8$^+$ T-cells from CMV+ individuals. Differences between CMV- and CMV+ individuals were tested by Mann-Whitney $U$ test. Stars indicate significant differences as follows: * $P$-value <0.05, ** $P$-value <0.01, *** $P$-value <0.001, *** $P$-value <0.0001. (TIF)

**S6 Fig. Fits of mathematical model to the deuterium enrichment data of CD4$^+$ T-cells.** Best fits of the kinetic heterogeneity model to the enrichment in true naïve (T$_{TN}$), central memory (T$_{CM}$), and effector memory (T$_{EM/EMRA}$) CD4$^+$ T-cells. Label enrichment in the DNA was scaled between 0 and 100% by normalizing for the maximum enrichment in granulocytes (see Materials and methods). The measurements that were out of range of the standards are indicated in red closed symbols, those in range are indicated in black closed symbols. E35, E30, E23, E21, and E10 are CMV- individuals, E59, E39, E34, E08, and E04 are CMV+ individuals. E30 only received $^2$H$_2$O for 3.5 weeks. (TIF)

**S7 Fig. Fits of mathematical model to the deuterium enrichment data of CD8$^+$ T-cells.** Best fits of the kinetic heterogeneity model to the enrichment in true naïve (T$_{TN}$), central memory

($T_{CM}$), effector memory ($T_{EM/EMRA}$), and CMV-specific CD8$^+$ T-cells. Label enrichment in the DNA was scaled between 0 and 100% by normalizing for the maximum enrichment in granulocytes (see Materials and methods). The measurements that were out of range of the standards are indicated in red closed symbols, those in range are indicated in black closed symbols. E35, E30, E23, E21, and E10 are CMV- individuals, E59, E39, E34, E08, and E04 are CMV+ individuals. E30 only received $^2$H$_2$O for 3.5 weeks.
(TIF)

**S8 Fig. Dynamics of different T-cell subpopulations and estimated production rates including confidence intervals. (A)** Combined deuterium labelling enrichment (%DNA labelled) of different T-cell subpopulations for all individuals, both CMV+ and CMV-. Label enrichment was scaled between 0 and 100% by normalizing for the estimated maximum enrichment of granulocytes (see Materials and methods). **(B)** Summary of estimated production rates of $T_{TN}$, $T_{CM}$, and $T_{EM/EMRA}$ CD4$^+$ and CD8$^+$ T-cells in all individuals. **(C)** Summary of estimated production rates with confidence intervals of $T_{TN}$, $T_{CM}$, and $T_{EM/EMRA}$ CD4$^+$ and CD8$^+$ T-cells in CMV- (blue symbols) and CMV+ (red symbols) individuals. Data from each individual are represented by unique symbols. **(D)** Summary of estimated production rates with confidence intervals of CMV-specific CD8$^+$ T-cells. All estimates in **(B)**, **(C)**, and **(D)** were obtained by fitting a single-exponential model to the data sets per individual (see Materials and methods and S6 Fig). E30 only received $^2$H$_2$O for 3.5 weeks. **(E)** Summary of estimated production rates of $T_{CM}$, $T_{EM/EMRA}$ and CMV-specific CD8$^+$ T-cells by fitting a multi-exponential model to the data sets per individual. Differences between groups were assessed by Kruskal-Wallis test. Stars indicate significant differences as follows: $^*$ $P$-value $<0.05$, $^{**}$ $P$-value $<0.01$, $^{***}$ $P$-value $<0.001$, $^{***}$ $P$-value $<0.0001$.
(TIF)

**S9 Fig. $^2$H enrichment in urine and granulocytes and parameter estimates from urine enrichment curves. (A)** Best fits of the mathematical model to the $^2$H enrichment in urine (see Materials and methods for differential equations). **(B)** Parameter estimates of the urine enrichment curves in all participants, where c represents the amplification factor, f the fraction of $^2$H$_2$O in the drinking water, δ the turnover rate of body water per day, and β the plasma enrichment due to the boost at day 0. **(C)** Best fits of enrichment curves in granulocytes (see Materials and methods). The measurements that were out of range of the standards are indicated in red closed symbols, those in range are indicated in black closed symbols. E35, E30, E23, E21, and E10 are CMV- individuals, E59, E39, E34, E08, and E04 are CMV+ individuals. E30 only received $^2$H$_2$O for 3.5 weeks.
(TIF)

**S1 Table. CMV-specific CD8$^+$ T-cell dextramer staining.** The selected individuals for the deuterated water labelling study are shaded in the table and the selected dextramer+ CMV-specific CD8$^+$ T-cell populations are bold and italic. Individuals were tested with dextramers that corresponded with their HLA-type. 0 = no response to dextramer; NA = not applicable.
(DOCX)

**S2 Table. Median (range) of loss rates ($d^*$) of labelled cells of different T-cell subsets.**
(DOCX)

## Acknowledgments

We thank Arpit Swain for useful discussions, René van Boxtel and Inge Pronk for help in designing the study, Eva Lukas for helping resolve a technical issue, and Jeroen van Velzen, Pien van der Burght and Sebastian van Burgh for help with the flow cytometry sorting.

## Author Contributions

**Conceptualization:** Sara P. H. van den Berg, Lyanne Y. Derksen, Rob J. de Boer, Kiki Tesselaar, José A. M. Borghans, Debbie van Baarle.

**Data curation:** Sara P. H. van den Berg, Lyanne Y. Derksen.

**Formal analysis:** Sara P. H. van den Berg, Lyanne Y. Derksen, Julia Drylewicz.

**Funding acquisition:** José A. M. Borghans, Debbie van Baarle.

**Investigation:** Sara P. H. van den Berg, Lyanne Y. Derksen, Julia Drylewicz, Nening M. Nanlohy, Lisa Beckers, Josien Lanfermeijer, Stephanie N. Gessel, Martijn Vos, Sigrid A. Otto.

**Methodology:** Sara P. H. van den Berg, Lyanne Y. Derksen, Julia Drylewicz, Nening M. Nanlohy, Lisa Beckers, Rob J. de Boer, Kiki Tesselaar, José A. M. Borghans, Debbie van Baarle.

**Project administration:** Sara P. H. van den Berg, Nening M. Nanlohy, Lisa Beckers.

**Resources:** Kiki Tesselaar, José A. M. Borghans, Debbie van Baarle.

**Supervision:** Sara P. H. van den Berg, Josien Lanfermeijer, Rob J. de Boer, Kiki Tesselaar, José A. M. Borghans, Debbie van Baarle.

**Validation:** Sara P. H. van den Berg, Lyanne Y. Derksen.

**Visualization:** Sara P. H. van den Berg, Lyanne Y. Derksen.

**Writing – original draft:** Sara P. H. van den Berg, Lyanne Y. Derksen.

**Writing – review & editing:** Sara P. H. van den Berg, Lyanne Y. Derksen, Julia Drylewicz, Rob J. de Boer, Kiki Tesselaar, José A. M. Borghans, Debbie van Baarle.

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
