## [Decision Letter · Decision Letter 0]

17 Jun 2021

Dear Ms. Derksen,

Thank you very much for submitting your manuscript "Quantification of T-cell dynamics during latent cytomegalovirus infection in humans" for consideration at PLOS Pathogens. As with all papers reviewed by the journal, your manuscript was reviewed by members of the editorial board and by several independent reviewers. In light of the reviews (below this email), we would like to invite the resubmission of a significantly-revised version that takes into account the reviewers' comments.

We cannot make any decision about publication until we have seen the revised manuscript and your response to the reviewers' comments. Your revised manuscript is also likely to be sent to reviewers for further evaluation.

Sincerely,

Laurent Coscoy

Associate Editor

PLOS Pathogens

Erik Flemington

Section Editor

PLOS Pathogens

Kasturi Haldar

Editor-in-Chief

PLOS Pathogens

orcid.org/0000-0001-5065-158X

Michael Malim

Editor-in-Chief

PLOS Pathogens

orcid.org/0000-0002-7699-2064

Reviewer's Responses to Questions

**Part I - Summary**

Reviewer #1: The manuscript by van den Berg et al. investigates the dynamic of T cells in older people latently infected with human cytomegalovirus (HCMV). The authors describe marker distribution on HCMV-specific and bulk T-cells 45 persons and analyzed the dynamics of T cell memory subsets for up to 500 days by in vivo deuterium labeling in a smaller proportion. Here, the authors showed that the dynamics of CMV-specific and bulk memory T cells were comparable in both populations and CMV-specific T cells have no longer life-span than the bulk memory T cell population.

The manuscript is well written and the authors discussed in detail their findings and the limitation of the study. The presented experimental data support the drawn conclusions, but remain mostly descriptive. Due to the high interest for immunologists and virologists in the phenomenons of memory inflation of CMV-specific cells and the potential role of CMV in immunosenescence the manuscript should be considered for acceptance in PLoS pathogens after targeting the following points.

Reviewer #2: I read with interest the paper “Quantification of T-cell dynamics during latent cytomegalovirus infection in humans” by van den Berg et al. The authors study T-cell dynamics in healthy CMV seropositive people and show that the CD4 compartment is affected by CMV latency more vigorously than the CD8 subset, that immunodominant CMV-specific CD8 cells possess an overall distinct phenotype than the general TEMRA or EM counterparts and that the populations of CMV specific cells are rather stable in the long run. The paper is generally well written, and the results well interpreted. The discussion is somewhat long, but it covers many aspects that need to be addressed to qualify the results and put them in context of the literature and of the scientific consensus on CMV immunity and aging.

Reviewer #3: This is an interesting study that addresses a range of features regarding CD4+ and CD8+ T cells in relation to CMV serostatus.

There are interesting results within the work but it will be important to bring out the novelty and limitations of the work.

A concern is that the clarity of the writing is somewhat vague and the scientific message is not clear. The title and abstract are somewhat reflective of this.

**Part II – Major Issues: Key Experiments Required for Acceptance**

Reviewer #1: 1. As far as I understand the authors the authors only used selected multimers specific for a defined combination of peptide and HLA-molecule to analyze the CMV-specific response in the donors. The multimers can detect epitopes that are immunodominant from the total population point of view. Nevertheless, it is known from the literature that the individual response against CMV is highly variable (for analyses see Sylwester et al. (already cited) Toya et al. PMID: 33335252, Attaf et al. PMID: 3310126, Gabel et al PMID: 3310126, Smith et al. PMID: 3310126) and therefore the author will miss most of the CMV-specific response in their analyses. They mentioned this problem in the discussion, but do not target it in their experiments. I was wondering why they do not try to get a better understanding of the total CMV response by using peptide cocktails or APCs loaded with cell extracts of infected cells to estimate the total response for each donor.

2. The authors selected for the in vivo labeling the 5 donors with the highest number of multimer-positive CD8 T cells, because they hypothesize that any difference would be more obvious in these donors. But it also could be that in these the differences might be less pronounced because here a steady-state situation has been already reached. When the authors are only interested in the maintenance of the memory pool it is fine to look at this situation. But when they also would be interested in the expansion of the memory pool (memory inflation), it would be better to include also donors with lower frequencies of epitope-specific T cells, because here the expansion of a limited number of T-cells clone might be better to observe.

Reviewer #2: Overall, the paper is highly interesting for experts studying CMV immunity and of interest for the more general audience of immunological gerontologists. However, some aspects need to be addressed before the paper may be published in PLOS Pathogens.

1. The authors show in Figures 1E and 1G MFI-based t-SNE cluster analyses that show distinct clustering of CD8 T cells recognizing CMV-specific immunodominant epitopes. This is highly interesting and relevant, but the paper would benefit from a more detailed representation of the markers showing differences between CMV-specific cells and general TEMRA or EM cells in two-dimensional representative dotplots.

2. The authors conclude that “the changes in the phenotype of the T-cell pool of CMV+ individuals cannot solely be explained by the presence of large numbers of CMV-specific T-cells.” They concede in the discussion that it cannot be excluded that CD8 recognizing subdominant CMV antigenic epitopes may have a phenotype that is closer to the general CD8 population, but dismiss the scenario, arguing that they used tetramers against 7 most abundant epitopes. This is an over interpretation, because they did not test if other HLA molecules that were present in their test subjects may carry additional antigenic epitopes. Furthermore, it is known that subdominant epitopes have phenotypes that are distinguishable from the dominant ones and in light of the huge genomic size of CMV it is impossible to exclude that numerous additional epitopes contributed to the TEMRA pool and formed it. Therefore, this argument needs to be altered in the final paragraph of the introduction and in the abstract, introducing the alternative scenario as delineated above.

3. In figure 3C, the authors show several comparisons and one that produced a p=0.095 in the CD4 TEM/EMRA fraction. It is not clear if they corrected the stat analysis for the number of tests performed, which would raise the p value and void this claim. If a Bonferroni-like correction was performed, the argument may stand, but if not, the authors are advised to avoid this claim. In any case, given the rather high level of the p values, the wording may benefit from a more neutral assessment (e.g.: Mathematical modelling supported a trend towards a lower production)

Reviewer #3: Major issues

Much is made of the statement: "The impact of CMV infection on the T-cell pool

stretches beyond the presence of large CMV-specific CD8+ T-cell populations". What does this mean ? Does it suggest that T cells specific for other antigens also have altered phenotype in CMV+ people? As the authors state, there will be large populations of CD8+ cells that are specific for CMV but are not detected by the dextramers. This could easily be the explanation. Did the authors consider the use of dextramers against other viral epitopes to test this theory? The statement appears overstated.

The novel aspects of the work need to be addressed.

There are very many papers on immunophenotyping of CMV-specific T cells and the tSNE plots do not add substantially. What are the major novel points?

The reduction of Ki-67 on CD4+ cells in CMV+ people seems novel and perhaps is a major point of the paper. Is there a hypothesis for this ?

I have concerns that the analysis is somewhat overstated. The p-value for reduction in average production/loss rate of CD4+ TEM/EMRA cells in CMV+CD4 proliferation is 0.1 but this association has a results paragraph and is commented on many times as 'trend', 'hint'. A one sentence note of the trend is all that is needed and the rest should be excluded.

In addition, the positive association of deuterium proliferation with Ki67 is only made by exclusion of one subject, which appears inappropriate.

**Part III – Minor Issues: Editorial and Data Presentation Modifications**

Reviewer #1: Minor points:

1. The rationale of analyzing memory dynamics only in older persons remains unclear.

2. The authors mentioned in the third paragraph of the discussion the role of reactivation for the establishment of the unique CMV-specific memory. The term reactivation implies for the broader audience production of viral progeny, but in the last years, it comes more clear that transcriptional activity of latently infected cells might be the reason for T cell expansion during viral latency. Please see work by the labs of Felicia Goodrum, Mark Wills, John Sinclair (all for HCMV), Matthias Reddehase (for MCMV).

3. In the same paragraph, the authors define “Memory Inflation” as an increase of memory T cells. Specifically for mice, it is clear that in animals with a high latent load (after high dose infection or after HCT) the inflationary cells are effector cells with some memory cell-like features (Baumann et al. PMID: 29652930), true Inflation of memory cells was only recently shown by Holtappels et al. PMID: 32707744.

4. In figure 4A short time increase and decrease of total T cells and frequency are visible in some of the donors. Do the authors have any explanation, why this is not reflected in the subfigure 4B.

5. In the discussion; I miss at some point the inclusion of the knowledge obtained from MCMV infected mice (4th paragraph, last two paragraphs).

Further comments

1. In the paragraph PBMC, neutrophilic granulocytes, and serum isolation and storage, please use molarity/concentration and not weight for the buffer composition

2. In some of the figures S5 & S6, it is hard to assign the donors to the CMV+ or CMV- group. Please make it more clear.

Reviewer #2: In general, the authors should adapt statements like “Based on t-SNE analysis, we found that CMV-specific CD8+ T-cells only partially explain the differences …” into something like “Based on t-SNE analysis, we found that IMMUNODOMINANT CMV-specific CD8+ T-cells only partially explain the differences …”

Reviewer #3: it might be helpful for the authors to consider what the take home messages are from the paper and the messages that would get this cited. That could help the writing style. Would a summary cartoon be useful ?

PLOS authors have the option to publish the peer review history of their article (what does this mean?). If published, this will include your full peer review and any attached files.

Reviewer #1: No

Reviewer #2: **Yes: **Luka Cicin-Sain

Reviewer #3: No
---

## [Decision Letter · Decision Letter 1]

20 Oct 2021

Dear Ms. Derksen,

Thank you very much for submitting your manuscript "Quantification of T-cell dynamics during latent cytomegalovirus infection in humans" for consideration at PLOS Pathogens. As with all papers reviewed by the journal, your manuscript was reviewed by members of the editorial board and by several independent reviewers. The reviewers appreciated the attention to an important topic. Based on the reviews, we are likely to accept this manuscript for publication, providing that you modify the manuscript according to the review recommendations of reviewer 2.

Sincerely,

Laurent Coscoy

Associate Editor

PLOS Pathogens

Erik Flemington

Section Editor

PLOS Pathogens

Kasturi Haldar

Editor-in-Chief

PLOS Pathogens

orcid.org/0000-0001-5065-158X

Michael Malim

Editor-in-Chief

PLOS Pathogens

orcid.org/0000-0002-7699-2064

Reviewer Comments (if any, and for reference):

Reviewer's Responses to Questions

**Part I - Summary**

Reviewer #1: In the revised version the authors carefully answered the points raised in my first statement. They used the time to enhance readability and make the take home message more clear.

Reviewer #2: The authors have addressed all of my concerns except the first major concern, on the tSNE analysis in Fig. 1E-1G.

The conversation was:

Q. The authors show in Figures 1E and 1G MFI-based t-SNE cluster analyses that show distinct clustering of CD8 T cells recognizing CMV-specific immunodominant epitopes. This is highly interesting and relevant, but the paper would benefit from a more detailed representation of the markers showing differences between CMV-specific cells and general TEMRA or EM cells in two-dimensional representative dotplots.

A. We have analyzed the differences between CMV-specific T-cells and bulk TEMRA, TEM and TCM cells using conventional gating in two-dimensional plots for the highly relevant markers CD28, CD57, and KLRG-1 (Figure 2D), which are often associated with CMV-seropositivity and indicative of their phenotype. We used multidimensional t-SNE analysis to investigate

specifically whether the differences between the T-cell pools of CMV- and CMV+ subjects could be explained by the sheer presence of large, immunodominant CMV-specific T-cell populations in CMV+ individuals. We feel that following up the multi-dimensional t-SNEanalysis by conventional flow cytometric gating in two dimensions, would in a way be a step backward, and that providing individual dot plots would have limited added value as this would rely on a few representative dot-plots.

I concede that the dot-plots may be a step back, and that the heatmaps provided in figure 1F and 1H provide a more detailed information on the make-up of the tSNE subsets than the representative dot-plots could. Nevertheless, the discussion reveals the problem that well-intentioned readers will have to comprehend the data provided by the non-intuitive tSNE plots. In light of that, it would help the readers (and the authors) if the paper would provide in a separate panel the information on the percentage of cells, within the tetramer labelled groups, that belong to each of the the nine tSNE clusters in the individual cell donors. This information will help us to assess if the information from the representative tSNE in 1G is reproduced in the majority of the individuals, or if variations are more common.

This is particularly important because the cluster 9 features a phenotype that is not typically associated with TEMRA cells in inflationary CD8 responses to CMV infection, yet seems to be the dominant one among the antigen specific cells in the representative tSNE in 1G.

Reviewer #3: The authors have made considerable changes to the manuscript and this has significantly improved the clarity and message

**Part II – Major Issues: Key Experiments Required for Acceptance**

Reviewer #1: does not apply

Reviewer #2: See above

Reviewer #3: Nil

**Part III – Minor Issues: Editorial and Data Presentation Modifications**

Reviewer #1: on page 20 the authors mentioned a manuscript as submitted, please check here if PLos allows this statement. If the authors what to hold the statement in the body of the text, I would recommend to publish the submitted manuscript as a Preprint and include it in the reference list.

Reviewer #2: None

Reviewer #3: Nil

PLOS authors have the option to publish the peer review history of their article (what does this mean?). If published, this will include your full peer review and any attached files.

Reviewer #1: **Yes: **Niels A. Lemmermann

Reviewer #2: No

Reviewer #3: No

Figure Files:

Data Requirements:

Reproducibility:

References:

---

## [Editor Report · Decision Letter 2]

27 Nov 2021

Dear Ms. Derksen,

We are pleased to inform you that your manuscript 'Quantification of T-cell dynamics during latent cytomegalovirus infection in humans' has been provisionally accepted for publication in PLOS Pathogens.

Best regards,

Laurent Coscoy

Associate Editor

PLOS Pathogens

Erik Flemington

Section Editor

PLOS Pathogens

Kasturi Haldar

Editor-in-Chief

PLOS Pathogens

orcid.org/0000-0001-5065-158X

Michael Malim

Editor-in-Chief

PLOS Pathogens

orcid.org/0000-0002-7699-2064
---

## [Editor Report · Acceptance letter]

10 Dec 2021

Dear Ms. Derksen,

We are delighted to inform you that your manuscript, "Quantification of T-cell dynamics during latent cytomegalovirus infection in humans," has been formally accepted for publication in PLOS Pathogens.

Best regards,

Kasturi Haldar

Editor-in-Chief

PLOS Pathogens

orcid.org/0000-0001-5065-158X

Michael Malim

Editor-in-Chief

PLOS Pathogens

orcid.org/0000-0002-7699-2064